# TOOL DECODING: A PLUG-AND-PLAY APPROACH TO ENHANCING LANGUAGE MODELS FOR TOOL USAGE

## ABSTRACT

Despite the significant advancements in large language models (LLMs), their tool-use capabilities remain limited. This limitation stems from the fact that existing approaches often merely adapt strategies designed for basic natural language tasks, overlooking the specific challenges inherent in tool usage, such as precise tool selection, strict predefined formats, and accurate parameter assignment. To bridge this gap, we conduct a fine-grained analysis of the tool usage process, breaking it down into three critical stages: tool awareness, tool selection, and tool call. Our analysis reveals that most failures stem from selection errors, format violations, and parameter mis-assignments. Building on these insights, we propose **Tool Decoding**, a novel, training-free approach that directly incorporates tool-specific information into the decoding process. Tool Decoding employs constrained decoding to ensure format correctness and eliminate hallucinations, while leveraging order consistency to improve parameter accuracy through structured sampling and a majority-voting mechanism. This approach effectively addresses many common tool-use errors in a plug-and-play manner, allowing for seamless generalization to new tools as long as they are accompanied by well-structured documentation to guide the decoding process. Experimental evaluations on benchmarks like API-Bank and BFCL V2 • Live show that Tool Decoding leads to significant improvements across a diverse set of more than 10 models, including both generalist and tool-finetuned models. Almost all models demonstrate performance gains exceeding 70% on both benchmarks. Among the 7B-level models, five outperform GPT-3.5 on key tasks, with two even surpassing GPT-4.

## 1 INTRODUCTION

Recent advancements in large language models (LLMs) have significantly expanded their applications beyond basic natural language processing (NLP) tasks to more complex and dynamic functionalities (Qian et al., 2024; Li et al., 2024; Lu et al., 2024a). There is growing interest in equipping LLMs with external tools, allowing them to perform tasks that extend beyond traditional language generation, such as interacting with APIs to retrieve information, control devices, or even make complex decisions (Schick et al., 2024; Qin et al., 2024; Yao et al., 2023). Improving the tool-use capabilities of LLMs has emerged as a critical area of development, with the potential to significantly enhance their utility in various real-world scenarios.

As exemplified in Figure 1, when faced with some complex tasks, tool-augmented language models initially attempt to complete the task using natural language. If unsolvable, they transit to the tool-usage mode, generating tool calls to query the tool server and subsequently leveraging the server's response to complete the task (Qin et al., 2024; Huang et al., 2024). Specifically, to integrate external tools into LLM workflows, each tool is assigned a unique name, and a predefined tool call format is established, typically structured as `[ToolName(key1=value1, key2=value2)]`(Schick et al., 2024; Li et al., 2023). The tool usage process of LLMs can be divided into three key steps: (1) **Tool Awareness**, where the model identifies the need for external tools to accomplish the task, signaled by outputting the character `[` to enter tool mode; (2) **Tool Selection**, in which the model selects the most appropriate tool by generating its specific name immediately after `[`; and (3) **Tool Call**, where the model provides the correct parameters and completes the tool call according to the predefined format, then waits for the tool server's response.

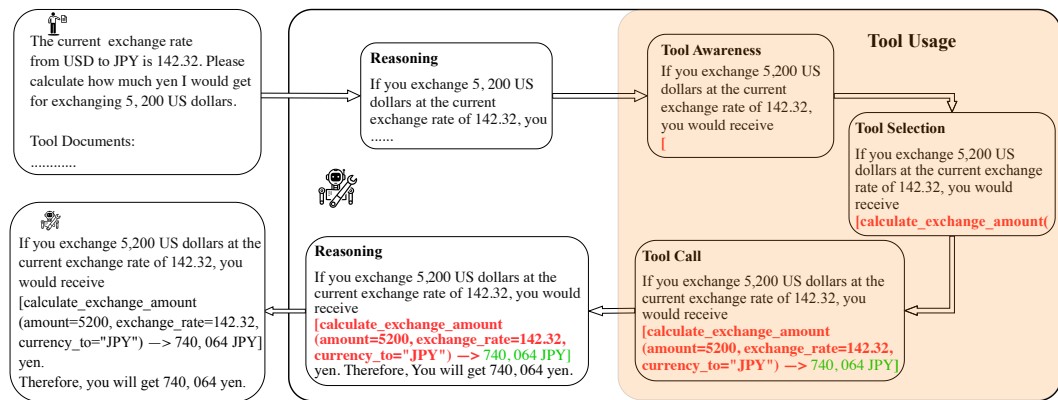

Figure 1: **The simplified workflow of tool-augmented language models.** During inference, tool-augmented language models continuously assess in real time whether tool usage is necessary. Once required, the model enters the tool-usage mode, selects the appropriate tool, and generates a tool call following the predefined format. The corresponding tool server detects the tool call, processes the request, and returns the response to the model, allowing it to proceed with the task.

| Error Type | | Example | Solution |
|---|---|---|---|
| Awareness | | You would receive 5200 yen.    no tool | – |
| Selection | | You would receive [currency_exchange_rate(currency_from='USD', currency_to='JPY') → 142.32] yen. | Constrained Decoding |
| Call | Format | You would receive [calculate_exchange_amount(amount=5200, exchange_rate=142.32, currency_to='JPY'] yen.    Missing a ) | |
| | Key | You would receive [calculate_exchange_amount(amount=5200, exchange_rate=142.32, currency_to='JPY'), from='USD')] yen. | |
| | Key | You would receive [calculate_exchange_amount(amount=5200, currency_to='JPY')] yen.    Missing required exchange_rate | Order Consistency |
| | Value | You would receive [calculate_exchange_amount(amount=200, exchange_rate=142.32, currency_to='JPY')] yen. | |

Table 1: Examples and our solutions for each error type across the three stages of tool usage.

While LLMs use tools by generating specific tokens, similar to basic NLP tasks, tool usage involves distinct characteristics, such as specific parameter requirements and the specialized structure of the tool call format. We notice that these unique features introduce a range of specific challenges, leading to most of the failure cases in practice. However, most prior research has neglected this aspect. Existing approaches can be broadly divided into two main categories: those based on supervised fine-tuning for task-specific tool usage (Schick et al., 2024; Patil et al., 2024; Qin et al., 2024), and those focusing on optimizing prompts for in-context learning by providing demonstrations (Yao et al., 2023; Liu et al., 2024b; Paranjape et al., 2023). These methods simply transfer approaches from basic NLP tasks without fully exploiting the unique potential inherent in tool usage.

In this work, we perform a fine-grained analysis of the tool usage process to explore the connection between failure cases and the unique characteristics of tool usage mentioned earlier. Based on the insights gained, we propose **Tool Decoding**, a novel plug-and-play method specifically designed to address the key challenges identified in this analysis without any additional training or fine-tuning. As illustrated in Figure 2, our analysis indicates that tool awareness is relatively straightforward and can be effectively handled even by less powerful models. However, tool selection and tool call are much more challenging due to their specific content and format requirements. Given the complexity of tool calls, we categorize call errors into three types: format errors, key errors, and value errors, as exemplified in Table 1. A comprehensive analysis of the five error types across all three stages reveals that tool usage failures are predominantly due to selection, format, and value

errors, as depicted in Figure 3. To address these issues, we propose Tool Decoding, which allows LLMs to better meet the specific requirements of tool usage while effectively addressing various types of errors. As shown in Table 1, constrained decoding is employed to eliminate format errors and reduce selection errors. while order consistency is applied to mitigate key and value errors. Tool Decoding combines these strategies, enabling models to accurately recognize and invoke tools without requiring additional training.

Tool Decoding is highly adaptable, as it does not depend on training data to learn tool interactions. Instead, it dynamically applies tool-related knowledge during the decoding stage, significantly improving the ability of a wide range of models to accurately select and invoke tools. Since no training is required, Tool Decoding can easily generalize to new tools, as long as they are accompanied by well-organized documentation to guide the decoding process. Moreover, Tool Decoding is also flexible to be combined with previous methods such as supervised fine-tuning, allowing for seamless integration and joint usage. By eliminating the need for extensive training or fine-tuning, our method offers a more efficient and flexible solution for enhancing LLM tool usage, making it suitable for a wide variety of models and scenarios.

In summary, the main contributions of this paper are:

- We conduct a fine-grained analysis of the three stages of the tool usage process and their associated errors, identifying the key bottlenecks in LLMs' tool usage capabilities.

- We propose Tool Decoding, a novel, training-free method that enhances tool usage in LLMs based on our analysis. This method leverages tool-specific information and structure during the decoding process to effectively address the primary errors in LLMs' tool usage.

- We validate Tool Decoding's superior performance by integrating it with a wide range of generalist and tool-finetuned models, evaluating them on the API-Bank[1] (Li et al., 2023) and BFCL V2 • Live[2]. Our experiments demonstrate that Tool Decoding significantly enhances performance across all models. Almost all models exhibit performance gains exceeding 70% across both benchmarks. Among the 7B-level models, five outperform GPT-3.5 on key tasks, and two even surpass GPT-4.

This work highlights the critical importance of tool-specific features and lays a foundation for future research aimed at improving LLMs' tool-use capabilities by exploiting these unique features. It also underscores the significant potential of decoding methods.

## 2 FINE-GRAINED ANALYSIS OF TOOL USAGE

In this section, we analyze the key challenges faced by LLMs in tool usage. By breaking down the tool usage process into distinct stages and performing detailed error analysis, we aim to pinpoint the primary bottlenecks and error patterns in the models' performance. This comprehensive evaluation sheds light on the unique demands of tool usage and provides insights into how LLMs can be improved to better handle these tasks.

**Analysis of Stages** To better understand the tool-use capabilities of LLMs, we divide the entire tool usage process into three stages: Tool Awareness, Tool Selection, and Tool Call. LLMs must successfully complete all three steps to use tools correctly. By evaluating the model's performance at each of these stages separately, we aim to identify the bottlenecks in its tool usage capabilities. To achieve this, we conduct detailed experiments for each stage using the Qwen1.5 models, across scales ranging from 1.8B to 72B parameters, within the UltraTool benchmark (Huang et al., 2024). For a detailed introduction to UltraTool, please refer to Appendix B.3. As illustrated in Figure 2, our analysis reveals that the difficulty of the tool usage process increases progressively across the three stages. While tool awareness is relatively straightforward and can be effectively managed even by small models, the challenges intensify in the tool selection and tool call stages due to specific content and format requirements. Notably, tool call presents the greatest complexity, with the best performing model achieving around 75% accuracy in this stage, underscoring the need for more targeted approaches to improve performance in this stage.

---

[1]https://github.com/AlibabaResearch/DAMO-ConvAI/tree/main/api-bank

[2]https://github.com/ShishirPatil/gorilla/tree/main/berkeley-function-call-leaderboard

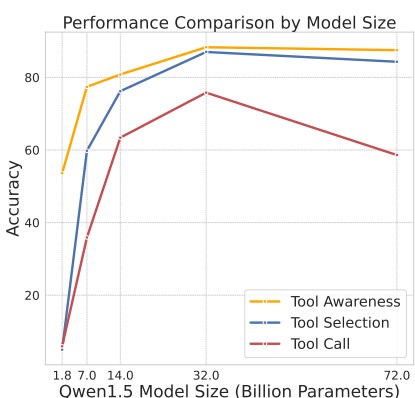

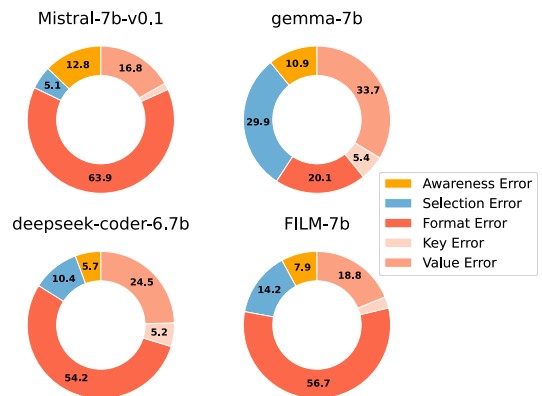

Figure 2: Performance comparison across the three stages of tool usage for a series of Qwen1.5 models, ranging in size from 1.8B to 72B, evaluated on the UltraTool dataset.

Figure 3: Proportion of error types for different LLMs on the API-Bank (Call) dataset (Li et al., 2023). The color schemes represent the tool-use stages corresponding to the error types.

**Analysis of Errors**    While some existing works have conducted coarse error analyses, their evaluations are not sufficiently comprehensive and lack a systematic approach. For instance, the analysis in API-Bank (Li et al., 2023) overlooks value errors and includes ambiguous error types such as `Has Exception`, limiting both clarity and utility. In contrast, we conduct a stage-specific and comprehensive error analysis, systematically identifying errors at each stage to derive fine-grained insights. Given the complexity of tool calls, we categorize call errors into three types: (1) Key Error: LLMs generate incorrect keys or the missing required parameter keys[3]; (2) Value Error: LLMs assign incorrect values to certain parameters; (3) Format Error: LLMs generate a tool call that does not adhere to the predefined format, rendering it undetectable by the tool server. Table 1 provides examples of each error type. As shown in Figure 3, errors in the tool call stage account for the highest proportion, followed by the tool selection stage, which is consistent with the experimental results presented in Figure 2. The Error type distribution of 70B-level models are exhibited in Figure 7, which is almost consistent with that of smaller models  Awareness errors account for only a small proportion and are almost impossible to improve through non-training methods, so we set them aside in this work. The most common errors, including selection errors, format errors, and value errors, arise from the specific format and functional demands of tool usage. Tool selection limits models to generate content within a specific range, while the format requirements of tool calls constrain models to adhere to a predefined structure. Additionally, the parameter values for tool calls require the model to fill them in sequentially, similar to completing a cloze test. These unique requirements highlight the mismatch between tool usage and standard language generation. By adjusting the decoding process of models to accommodate these specific requirements, these issues can be alleviated.

## 3 TOOL DECODING

In Section 2, we demonstrate that the main causes of failure in tool usage include selection errors, format errors, and value errors. To minimize these errors, we introduce Tool Decoding, a novel plug-and-play method that integrates tool-specific information into the LLMs' decoding process, as illustrated in Figure 4. This approach efficiently enhances the tool-use capabilities of LLMs by providing comprehensive support for both the tool selection and the tool call stages. Tool Decoding consists of two key components: constrained decoding and order consistency. The details of constrained decoding are discussed in subsection 3.1. For tool selection, constrained decoding is applied to restrict candidate tokens to valid tool names, preventing model hallucinations. During the tool call stage, it ensures the correctness of both the tool call format and optional parameter keys.

---

[3]Tool parameters are typically divided into required parameters and optional parameters, which are usually presented in a well-structured format in the tool documentation.

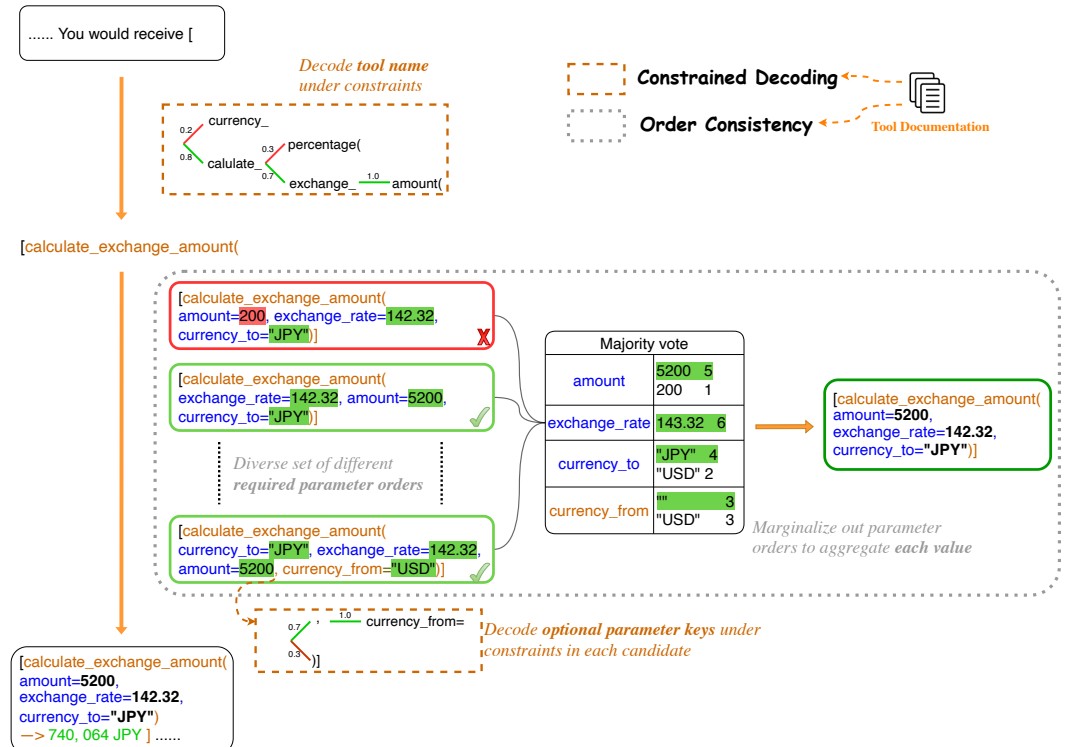

Figure 4: **Illustration of the Tool Decoding process.** The model is provided with a tool-use task description and a set of candidate tools, along with their respective documentation. Once the model recognizes the need for tool usage, the Tool Decoding method is invoked. Constrained decoding is applied to generate the tool name and optional parameter keys, while order consistency improves the accuracy of each parameter value. In the model's response, black text represents content generated through regular decoding, brown text indicates content generated through constrained decoding, and blue text highlights required parameter keys directly supplied to guide different orders.

Order consistency, detailed in subsection 3.2, is used to sample multiple candidate values for each parameter, with majority voting employed to eliminate key errors and minimize value errors.

## 3.1 CONSTRAINED DECODING

Figure 4 illustrates the complete Tool Decoding process, with constrained decoding as a crucial part of this method. If the model generates by regular decoding, the unrestricted vocabulary space could potentially result in an incorrect format or a non-existent tool name. To address these issues, our approach restricts the model to consider only a specific set of tokens, guided by the constraints imposed by the predefined format and tool information.

Since tool documentation is typically well-structured, extracting constraints using simple rules is feasible. We use regular expressions to extract each tool name, along with its required and optional parameters, and store these constraints in a lookup dictionary. During the inference stage, we query the lookup dictionary to retrieve the relevant constraints as the model begins generating the tool name and optional parameter keys. These constraints are then used to restrict the vocabulary space at each step until this portion is completed.

For example, as shown in Figure 4, when the model generates [ and enters the tool selection stage, all tool names are retrieved from the lookup dictionary and tokenized into a constrained token tree, along with the corresponding format element (. In the subsequent steps, the model is constrained to decode within the subtree of the current node at each step, ensuring that the vocabulary space is limited to the child nodes, which guarantees that the generated tool name is one of the provided tools. Similarly, after all required parameters have been assigned, the model may either use some optional

| model | vanilla | reverse | shuffle | aggregation |
|---|---|---|---|---|
| mistral-7b-v0.1 | 63.7 | 63.2 | 62.7 | **63.9** |
| FILM-7b | 68.4 | 65.4 | 69.7 | **69.7** |
| deepseek-coder-6.7b | 68.9 | 70.2 | 69.4 | **70.7** |
| xLAM-7b-r | 70.2 | 72.5 | 69.9 | **72.5** |

Table 2: Accuracy (%) of various models on the API-Bank (Call) dataset under different required parameter orders. The *aggregation* column shows the accuracy after applying majority voting across the results of the three orders. Underlined results indicate the best performance for each model across different parameter orders.

parameters or terminate parameter assignment with a closing parenthesis ) . Therefore, we construct a constrained token tree using all unused optional parameter keys and the closing parenthesis ) .

## 3.2 ORDER CONSISTENCY

While constrained decoding effectively eliminates format errors and mitigates selection errors, it falls short in addressing value errors. To address this, we introduce order consistency, which fully utilizes the property that tool calls remain functionally equivalent as long as the parameter values are consistent, regardless of the order in which the parameters are provided. By guiding the model to assign parameters in different orders, we can generate multiple tool calls for the same scenario and then apply majority voting to identify the most consistent value for each parameter. This method improves overall accuracy by reducing value errors and ensuring robustness across different parameter configurations. Our method is inspired by self-consistency, which improves reasoning ability of LLMs by generating multiple answers via different reasoning paths and then aggregating them, but overcomes the barrier to apply its thought to tool usage due to the absence of reasoning process.

In Table 2, we evaluate the accuracy on the API-Bank (Call) dataset across three generalist models and one tool-finetuned model under different parameter orders. The results indicate that changing the parameter order only leads to slight variations in the models' performance when generating tool calls. In some cases, using a parameter order different from that specified in the tool documentation even improves the model's performance. However, no single order proves to be universally superior, while the aggregated results from all orders surpass even the best individual order. This preliminary experiment highlights the need for introducing order consistency to further enhance overall performance.

Specifically, we fetch the required parameter keys from the lookup dictionary and shuffle them. As illustrated in Figure 4, after the model finishes generating the tool name, we sequentially append the parameter keys to the input, guiding the model to generate the corresponding values one by one. Once all required parameter values are generated, constrained decoding is applied to allow the model to determine whether any optional parameters are needed. Note that the transition between two parameters is triggered when the previous value is detected as fully generated. Consequently, we can obtain a set of candidate values for each parameter by sampling tool calls with different required parameter orders and retaining only those that meet the parameter type requirements. We then aggregate the tool calls by marginalizing over the orders and selecting the most consistent value for each parameter across the generated tool calls. Finally, the tool call derived from majority voting is used to request a response from the tool server. This method not only reduces value errors but also ensures the completeness of required parameters, preventing issues such as missing keys. For tools with many required parameters, there can be multiple parameter orders. We set an upper limit for the number of sampled tool calls, denoted as $oc$, with $oc \leq 12$ unless otherwise specified.

## 4 EXPERIMENTS

### 4.1 SETUP

**Tasks and Datasets** We evaluate Tool Decoding on the following benchmarks.

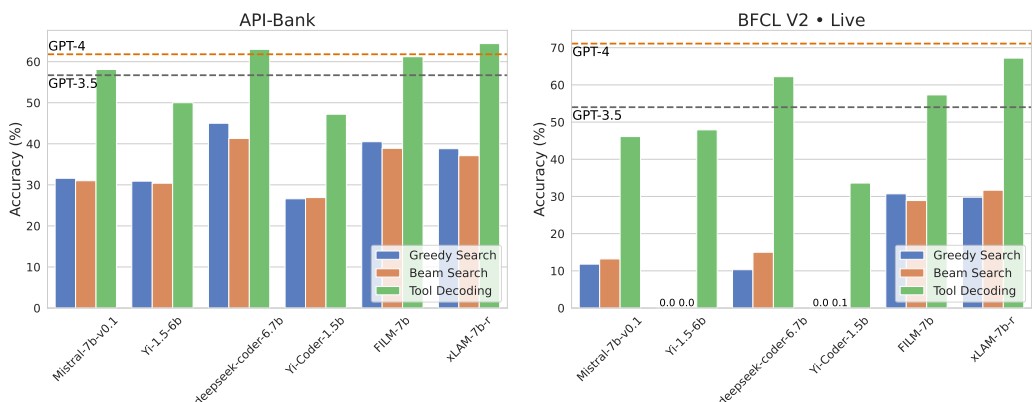

Figure 5: Total accuracy on the API-Bank and BFCL V2 • Live datasets, comparing Tool Decoding with greedy search and beam search across five generalist models and one tool-finetuned model. Additional results and evaluations on a broader range of models can be found in Appendix G.

| Model | Decoding Method | ICL Example Numbers | | | | |
|---|---|---|---|---|---|---|
| | | **0** | **2** | **4** | **6** | **8** |
| GPT-4 | Greedy Search | 76.2 | 72.7 | 72.2 | 73.7 | 73.4 |
| Mistral-7b-v0.1 | Greedy Search | 31.3 | 47.1 | 45.1 | 50.4 | 43.6 |
| | Tool Decoding | 65.7 | 70.2 | 69.2 | 70.5 | 70.9 |
| deepseek-coder-6.7b | Greedy Search | 46.9 | 66.7 | 69.2 | 69.2 | 70.2 |
| | Tool Decoding | 70.9 | **74.4** | **76.7** | **76.9** | **77.4** |

Table 3: Performance comparison of different decoding methods across varying numbers of in-context examples on API-Bank (Call). **Bold** highlights the results that surpass GPT-4 under the same prompt settings.

- **Tool-use dialogues.** An important task for tool-augmented language models is to function as tool-enabled chatbots, capable of solving more complex user problems and addressing advanced needs. Therefore, API-Bank (Li et al., 2023), a widely used benchmark for tool-use dialogues, is well-suited for evaluating our method. API-Bank comprises three evaluation categories: Call, Retrieve+Call, and Plan+Retrieve+Call. Since the third category primarily evaluates the model's planning capabilities and is unrelated to tool usage, we concentrate on the first two categories. For detailed information on API-Bank, please refer to Appendix B.1.

- **Tool usage.** Our method is specifically designed to enhance the tool-use capabilities of tool-augmented language models, particularly in the stages of tool selection and tool call. To evaluate its effectiveness, we assess performance on the BFCL V2 • Live, focusing on improvements in these two stages. The BFCL V2 • Live consists of six evaluation categories, with Relevance and Irrelevance primarily evaluating the model's tool awareness, which is not the main focus of our study. Therefore, we concentrate on the other four categories for our evaluation. For further details on the BFCL V2 • Live, please refer to Appendix B.2.

**Base LLMs**    We evaluate Tool Decoding across a diverse set of models, including chat models, long-context models, code models, and lightweight models. Additionally, we assess its performance on two tool-finetuned models: xLAM-7b-r (Zhang et al., 2024) and Toolformer (Schick et al., 2024). A detailed introduction to these models is provided in Appendix A.

## 4.2 MAIN RESULTS

We present the results for six models in Figure 5, with additional details and results for more models provided in Appendix G. The first chart displays the results from API-Bank, and the second chart shows the results from the BFCL V2 • Live. Each model is assessed using three decoding methods: Tool Decoding, greedy search, and beam search. The bars in the charts represent the total accuracy achieved by each model on the corresponding benchmark.

**API-Bank** The results, as shown in the left chart of Figure 5, indicate that Tool Decoding significantly improves the performance of all models in tool-use dialogues, enabling them to better leverage tools for executing user instructions. The method demonstrates strong performance across various model types and serves as a valuable complement to tool-finetuned models. Notably, when integrated with Tool Decoding, some models, such as deepseek-coder-6.7b-base and xLAM-7b-r, even outperform GPT-4 in this benchmark. As a plug-and-play method, Tool Decoding can integrate seamlessly with prompt engineering. Table 3 presents the performance of different numbers of in-context learning (ICL) examples on API-Bank (Call). The results demonstrate that our method effectively combines with prompt engineering, significantly enhancing the model's tool usage capabilities. Notably, this combination even enables a 7B-level generalist model, deepseek-coder-6.7b, to surpass GPT-4 under the same prompt settings.

**BFCL V2 • Live** The right chart of Figure 5 presents the results. Tool Decoding consistently enhances the tool-use capabilities of all models, with total accuracy more than doubling compared to greedy search and beam search. Notably, even weaker models like Yi-1.5-6b and Yi-Coder-1.5b, which fail on nearly all test cases with greedy search and beam search, achieve significant improvements with Tool Decoding. Furthermore, the tool-finetuned model xLAM-7b-r, when combined with Tool Decoding, surpasses GPT-3.5 and approaches GPT-4 levels of performance. Similarly, other models such as deepseek-coder-6.7b-base and FILM-7b outperform GPT-3.5 on this benchmark.

Tool Decoding enables seamless integration with various existing approaches and models. Appendix E presents results obtained using our method with larger generalist models and more advanced tool-finetuned models. Appendix F compares our approach with other decoding methods for tool usage, while Appendix C provides a latency analysis, showcasing the computational efficiency of our method.

## 4.3 ERROR ANALYSIS AND ABLATION STUDY

In this subsection, we perform an analysis of the reduction in different error types after applying Tool Decoding, assessing the effectiveness of the method in addressing each error category. Furthermore, we conduct ablation studies to specifically assess the role of order consistency in mitigating value errors.

**Error Analysis** We perform a fine-grained analysis of several representative models using both greedy search and Tool Decoding. The results, presented in Figure 6, show the performance with greedy search in the first row and Tool Decoding in the second row. The comparison reveals that Tool Decoding almost entirely eliminates format and key errors, while significantly reducing selection errors. However, there is a slight increase in value errors, which arises because the resolution of format, key, and selection errors uncovers underlying value errors that were previously masked by these other issues.

**Ablation Study on Order Consistency** To evaluate the effect of order consistency on reducing value errors, we conduct comparative experiments using different $oc$ upper limits, which control the number of sampled tool calls. The results are presented in Table 4. We first record the number of value errors for each model when using Tool Decoding without order consistency ($oc \leq 1$), and then compare the reduction in value errors across different $oc$ limits. Each row in Table 4 displays the proportion of value error reductions for each model at the corresponding $oc$ limit. The results indicate a positive correlation between the reduction in value errors and the number of tool calls allowed with different parameter orders, thereby confirming the effectiveness of order consistency.

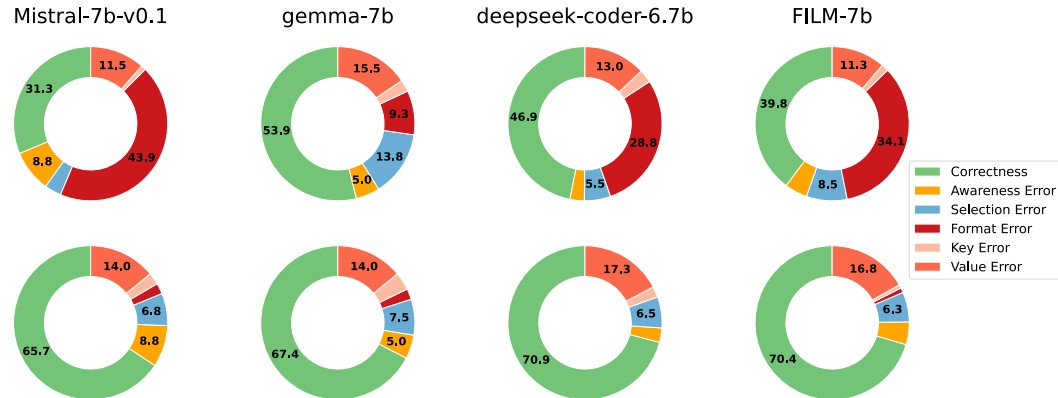

Figure 6: Error type distribution comparison of four LLMs on the API-Bank (Call) dataset, with and without Tool Decoding. The first row shows the results by greedy search, while the second row presents the results by Tool Decoding.

| Order Samples | Mistral-7b-v0.1 | gemma-7b | deepseek-coder-6.7b | FILM-7b |
|---|---|---|---|---|
| $oc \leq 4$ | 4.7 | 4.8 | 0.0 | 5.3 |
| $oc \leq 9$ | 7.8 | 8.1 | **9.3** | 2.7 |
| $oc \leq 12$ | **12.5** | **9.7** | 8.0 | **6.7** |

Table 4: Proportion (%) of value error reduction across 4 models on API-Bank when applying Tool Decoding with varying $oc$ limits (the upper limits for order samples) for order consistency, compared to results without order consistency ($oc \leq 1$).

## 5 RELATED WORK

**Tool-Augmented Language Models** Language models are constrained by the knowledge within their training data, limiting the range of tasks they can handle independently. For tasks involving numerical calculations, real-time information, or device control, models cannot perform autonomously and must rely on external tools (Feng et al., 2024; Shen et al., 2024; Li et al., 2024). Qin et al. (2024) describes the workflow of tool-augmented language models as a multi-step process: first, decompose the task and create a plan, which may be adjusted based on environmental feedback; second, use appropriate tools for each subtask; and finally, solve each subtask with the tool responses. Figure 1 provides a simplified example of this process. Various efforts aim to enhance the capability of LLMs with external tools. Yao et al. (2023); Liu et al. (2024b); Paranjape et al. (2023) utilize prompt engineering to enable models to interact with tools, but the effectiveness of this approach is limited by the model's inherent capabilities. Schick et al. (2024); Yang et al. (2024b); Tang et al. (2023); Liu et al. (2023); Qin et al. (2024); Li et al. (2023); Lu et al. (2024b) develop tool-augmented datasets to fine-tune models, enhancing their overall performance. While effective, this method is resource-intensive and lacks the flexibility to generalize to new tools.

**Tool Usage for Language Models** Tool-augmented language models need to manage the entire process of planning, tool usage, and response analysis. Some studies focus specifically on the tool usage step, also referred to as function calling. These studies require the model to generate appropriate tool calls directly, treating this as the task itself, rather than depending on external tools to complete additional tasks. In detail, Liu et al. (2024a); Xu et al. (2024); Du et al. (2024); Chen et al. (2024) propose novel methods for retrieving tools from large-scale tool libraries. Patil et al. (2024); Liu et al. (2024c); Mok et al. (2024) construct high-quality tool-use datasets to fine-tune models. However, these approaches merely adapt methods from basic NLP tasks to tool usage, which limits their ability to adequately address the specific demands of tool use or generalize effectively to new tools. Zhang et al. (2023); Wang et al. (2023a) introduce constrained decoding to enforce tool syntax in LLMs. While these methods reduce syntax errors, they do not address issues such as incorrect

parameter values. Wang et al. (2024) propose reranking to tackle this problem, but it requires training an additional scorer. In contrast, our Tool Decoding method mitigates various potential errors in tool usage without requiring any training.

**Sampling and Decoding in Language Models** A variety of decoding strategies have been proposed to improve language model performance, including top-$k$ sampling (Fan et al., 2018; Holtzman et al., 2018), temperature-based sampling (Ficler & Goldberg, 2017), and nucleus sampling (Holtzman et al., 2020). Beyond these, more refined algorithms have been developed to enhance reasoning capabilities. Wang et al. (2023b); Wang & Zhou (2024) introduce self-consistency as a method to improve Chain-of-Thought (CoT) reasoning by generating multiple candidate answers via different reasoning paths and aggregating them using majority voting. This approach enhances both the accuracy and robustness of reasoning tasks. Constrained decoding (Willard & Louf, 2023; Chen et al., 2022; Fang et al., 2023; Lu et al., 2022) improves generation quality by limiting the vocabulary to a smaller set of candidate tokens, effectively reducing the risk of hallucinations. While these methods have proven effective for basic NLP tasks, they are not directly applicable to tool usage. Our work bridges this gap by integrating these techniques with the unique features of tool usage.

## 6 Conclusion and Discussion

This paper introduces Tool Decoding, a training-free method that enhances LLMs' tool-use capabilities. A fine-grained analysis of tool usage reveals key errors in tool awareness, selection, and call stages, with most issues arising from incorrect tool selection, non-compliant format , and erroneous parameter assignments. Tool Decoding addresses these challenges by employing constrained decoding to ensure format correctness and leveraging order consistency to enhance the value accuracy of each parameter through majority voting. Experiments on API-Bank and BFCL V2 • Live show that Tool Decoding significantly boosts tool-use performance, with improvements exceeding 200% in some cases, enabling open-source models to match even surpass GPT-4 without training.

Looking ahead, Tool Decoding holds potential to improve the pass rate in tool-augmented dataset construction by ensuring accurate tool calls, thereby facilitating more efficient generation of fine-tuning data (Liu et al., 2024c). Its adaptability to new tools without the need for retraining makes it particularly valuable in dynamic, resource-constrained environments, opening the door to broader applications in both research and real-world scenarios.

## 7 Ethics Statement

This work does not involve any direct ethical concerns, as it focuses on developing a method for improving tool usage in large language models (LLMs) without introducing new ethical challenges. However, the widespread deployment of LLMs with enhanced tool-use capabilities could have implications for automation and human interaction with AI systems. It is important to consider the potential biases in the tools or data being used, as well as ensuring that LLMs are transparent in their decision-making processes. Developers should also be cautious in applying these systems to sensitive areas such as medical diagnosis or legal advice, where the accuracy and reliability of the model are critical. Furthermore, as our method does not require additional training, it offers an energy-efficient alternative to fine-tuning, reducing the carbon footprint associated with training large-scale models. However, we encourage continuous monitoring and evaluation of AI deployments to prevent unintended consequences and ensure they align with ethical guidelines.

## 8 Reproducibility Statement

To ensure the reproducibility of our results, we provide detailed explanations of our methods and experiments in the main paper and the appendix. In Section 2, we break down the tool usage process and error types, while Section 3 introduces the Tool Decoding method. Furthermore, we include comprehensive experimental setups in Section 4 and detailed benchmark descriptions in Appendix B.1 and B.2. In Appendix A, we provide information on the models and their configurations used in the experiments. All code and data used in the experiments will be released upon acceptance to ensure full transparency and reproducibility of our results.

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

## A    DETAILS FOR MODELS

- **GPT-4** (Achiam et al., 2023): GPT-4, developed by OpenAI, is a large-scale multimodal model capable of processing both text and images. It is more reliable, creative, and nuanced in its responses than its predecessors like GPT-3.5. In our experiments, we employ the gpt-4 (1106-Preview) version.

- **GPT-3.5** (Ye et al., 2023): This model is an improvement over GPT-3, focusing on reducing the hallucinations and factual errors present in GPT-3. It serves as the backbone for Chat-GPT and other similar applications. While it lacks the multimodal capabilities of GPT-4, it is still widely used for text-based tasks. In our experiments, we employ the gpt-35-turbo (0613) version.

- **Mistral-7B-v0.1** (Jiang et al., 2023): The Mistral-7B-v0.1 Large Language Model (LLM) is a pretrained generative text model with 7 billion parameters, developed by Mistral AI.

- **FILM-7B** (An et al., 2024): FILM-7B is a 32K-context LLM that overcomes the lost-in-the-middle problem. It is trained from Mistral-7B-Instruct-v0.2 by applying Information-Intensie (In2) Training.

- **deepseek-coder-6.7B-base** (Guo et al., 2024): Deepseek Coder is composed of a series of code language models, each trained from scratch on 2T tokens, with a composition of 87% code and 13% natural language in both English and Chinese.

- **gemma-7B** (Team et al., 2024): Gemma is a family of lightweight, state-of-the-art open models from Google, built from the same research and technology used to create the Gemini models.

- **Llama3-8B** (Dubey et al., 2024): Llama3-8B is part of Meta's LLaMA series, a family of models focused on providing a low-resource alternative to the more resource-intensive GPT models. It strikes a balance between efficiency and accuracy for text-based AI applications.

- **Qwen2-7B** (Yang et al., 2024a): Qwen2-7B is developed for Chinese and multilingual text processing. It is optimized for high-performance language understanding across various languages, making it a versatile choice for global applications.

- **Yi-1.5-6B** & **Yi-1.5-6B-Chat** (Young et al., 2024): Yi-1.5 is an upgraded version of Yi. It is continuously pre-trained on Yi with a high-quality corpus of 500B tokens and fine-tuned on 3M diverse fine-tuning samples. Compared with Yi, Yi-1.5 delivers stronger performance in coding, math, reasoning, and instruction-following capability, while still maintaining excellent capabilities in language understanding, commonsense reasoning, and reading comprehension.

- **Yi-Coder-1.5B** (Young et al., 2024):: Yi-Coder is a series of open-source code language models that delivers state-of-the-art coding performance.

- **Yi-1.5-34B** (Young et al., 2024): Yi-1.5 is an upgraded version of the Yi model family, with Yi-1.5-34B being the largest and most advanced model in this series.

- **deepseek-coder-33b** (Guo et al., 2024) Deepseek Coder is composed of a series of code language models with deepseek-coder-33b being the largest and most advanced model in this series.

- **gorilla-openfunctions-v2** (Patil et al., 2024) gorilla-openfunctions-v2 is one of the most powerful 7B-level tool-finetuned models.

- **xLAM-7b-r** (Zhang et al., 2024): Large Action Models (LAMs) are advanced large language models designed to enhance decision-making and translate user intentions into executable actions that interact with the world. LAMs autonomously plan and execute tasks to achieve specific goals, serving as the brains of AI agents.

- **Toolformer** (Schick et al., 2024): Toolformer is a specialized language model that can select and interact with external tools dynamically during inference, enhancing its ability to solve real-world problems without the need for retraining.

## B  DETAILS FOR DATASETS

### B.1  API-BANK

API-Bank is one of the pioneering benchmarks for tool-augmented LLMs, consisting of 2,202 dialogues involving 2,211 APIs across 1,008 domains. The dataset includes 934 dialogues in the Call category, 769 in the Retrieve+Call category, and 499 in the Plan+Retrieve+Call category. On average, each dialogue contains 2.76 turns in the training set and 2.91 turns in the testing set. Since the Plan+Retrieve+Call category primarily evaluates a model's planning capabilities, which is not our focus, we limit our experiments to the Call and Retrieve+Call categories.

Example B.3 shows a query and the corresponding prompt in API-Bank.

### B.2  BFCL V2 • LIVE

BFCL V2 • Live is a dataset designed to evaluate the function-calling (tool-use) capabilities of LLMs. It leverages live, user-contributed function documentation and queries, addressing issues of dataset contamination and biased benchmarks. By incorporating user-provided data, BFCL V2 • Live aims to more accurately assess LLM function-calling performance in real-world scenarios, highlighting the importance of models performing effectively in diverse and dynamic environments. The dataset comprises 258 simple, 7 multiple, 16 parallel, 24 parallel multiple, 875 irrelevance detection, and 41 relevance detection entries. Each test category is outlined in the Evaluation Categories section, providing a comprehensive assessment of various function-calling scenarios. Since irrelevance and relevance detection focus on tool awareness, which is not central to our work, we focus our experiments on the first four categories.

- **Simple Function:** Single function evaluation contains the simplest but most commonly seen format, where the user supplies a single JSON function document, with one and only one function call will be invoked.

- **Multiple Function:** Multiple function category contains a user question that only invokes one function call out of 2 to 4 JSON function documentations. The model needs to be capable of selecting the best function to invoke according to user-provided context.

- **Parallel Function:** Parallel function is defined as invoking multiple function calls in parallel with one user query. The model needs to digest how many function calls need to be made and the question to model can be a single sentence or multiple sentence.

- **Parallel Multiple Function:** Parallel Multiple function is the combination of parallel function and multiple function. In other words, the model is provided with multiple function documentation, and each of the corresponding function calls will be invoked zero or more times.

Example B.3 presents a query and its corresponding prompt from the parallel-multiple category of BFCL V2 • Live. The structure of other categories is similar to this.

### B.3  ULTRATOOL

UltraTool is a comprehensive benchmark designed to evaluate the ability of LLMs to effectively utilize tools in real-world scenarios. It focuses on the entire workflow of tool-augmented language models, covering each stage from initial planning and tool creation to their application in complex tasks. The benchmark provides a rich dataset that supports fine-grained analysis of each stage in this workflow, allowing for a deeper understanding of how models perform in various aspects of tool usage.

In our analysis, we focus on three key stages of tool usage—Tool Awareness, Tool Selection, and Tool Call—using the data provided by UltraTool, as shown in Figure 2.

**Dataset Example of API-Bank**

Based on the given API description and the existing conversation history 1..t, please generate the API request that the AI should call in step t+1 and output it in the format of [ApiName(key1='value1', key2='value2', ...)], replace the ApiName with the actual API name, and replace the key and value with the actual parameters. Your output should start with a square bracket "[" and end with a square bracket "]". Do not output any other explanation or prompt or the result of the API call in your output. This year is 2023.

**Input Template:**
User: [User's utterence]]
AI: [AI's utterence]

**Expected output:**
API: [ApiName(key1='value1', key2='value2', ...)]

**API descriptions:**

```
{
  "name": "GetUserToken",
  "description":......,
  "input_parameters": {
    "username": {
      "type": "str",
      "description": "The username of the user."
    },
    "password": {
      "type": "str",
      "description": "The password of the user."
    }
  },
  "output_parameters": {
    "token": {
      "type": "str",
      "description": "The token of the user."
    }
  }
  ...........
}
```

**Input:**
User: Can you add a schedule for me at 2pm on September 12th called "Meeting with John" at the office?
AI: Sure, I can add that schedule for you. When would you like the alarm to remind you?
User: Can you remind me 10 minutes before the schedule?
AI: Absolutely. To schedule the meeting, I first need to authenticate your account. Please provide your username, email, and password.
User: My username is JaneSmith, my email is janesmith@example.com, and my password is password.
AI: Thank you.

---

**Dataset Example of the Parallel-Multiple Category of BFCL V2 • Live**

## system:
You are an expert in composing functions. You are given a question and a set of possible functions. Based on the question, you will need to make one or more function/tool calls to achieve the purpose.
If none of the function can be used, point it out. If the given question lacks the parameters required by the function, also point it out. You should only return the function call in tools call sections.
If you decide to invoke any of the function(s), you MUST put it in the format of [func_name1(params_name1=params_value1, params_name2=params_value2...), func_name2(params)]
You SHOULD NOT include any other text in the response.

Here is a list of functions in JSON format that you can invoke.

```
[
    {
        "name": "view_service_provider_profile",
        "description": ......,
        "parameters": {
          "type": "dict",
          "required": ["professional_id"],
          "properties": {
            "professional_id": {
              "type": "integer",
              "description": ......,
            }
          }
        }
        ...........
    }
]
```

## user:
I need to find a maid for cleaning services who is available on March 19, 2024, starting at noon. Can you find someone with good ratings, maybe around 4 or 5 stars, and no record of quality problems?

---

## C LATENCY ANALYSIS

We conduct a latency analysis for Tool Decoding in comparison with greedy search and beam search on the API-Bank (Call) dataset, as shown in Table 5. The results show that Tool Decoding with $oc \leq 1$ introduces only slight latency, while Tool Decoding with $oc \leq 6$ is faster than beam search with the same number of samples. This is because order consistency maintains multiple candidates only during the generation of the tool call, which constitutes just a portion of the entire response. Note

| Decoding Method | Mistral-7b-v0.1 | deepseek-coder-6.7b |
|---|---|---|
| Greedy Search | 7.92 | 7.69 |
| Tool Decoding ($oc \leq 1$) | 8.57 | 8.41 |
| Beam Search ($beam = 6$) | 13.82 | 15.25 |
| Tool Decoding ($oc \leq 6$) | 9.74 | 10.7 |

Table 5: Inference speed (sec/sample) for Tool Decoding in comparison with greedy search and beam search.

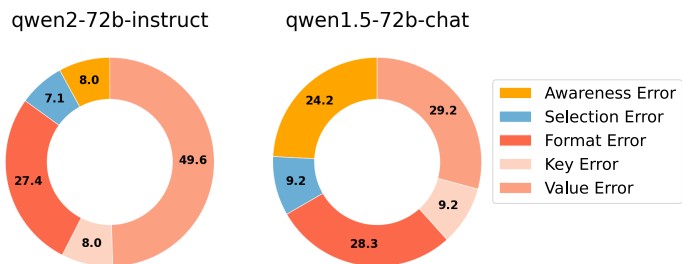

Figure 7: Error type distribution of two 70B-level LLMs on the API-Bank (Call) dataset. The accuracy of qwen2-72b-instruct is 71.4%, while qwen1.5-72b-chat achieves 69.9%.

| Model | Greedy Search | Tool Decoding | TOOLDEC |
|---|---|---|---|
| gorilla-openfunctions-v2 | 51.9 | **77.2** | 69.4 |
| Toolformer | 13.5 | **31.8** | 7.7 |
| deepseek-coder-6.7b | 46.9 | **70.9** | 65.7 |

Table 6: Performance comparison across different models using various decoding methods on API-Bank (Call). **Bold** highlights the best results for each model across the different decoding methods. The results demonstrate the advantages of our method over TOOLDEC across a range of tool-finetuned and code models.

that our current implementation applies constraints at the level of logits. For practical deployment, these constraints could be implemented at the language head layer, which would further reduce computational requirements and enhance processing speed.

## D  ADDITIONAL ERROR ANALYSIS

As shown in Figure 7, the error type distributions of the two 70B-level models share similar features with smaller models. Format errors and value errors remain the most prevalent, underscoring the challenges arising from the tool call stage.

## E  COMBINE WITH MORE POWERFUL MODELS

To provide a more comprehensive evaluation, we supplement our experiments with results on more powerful models, as shown in Table 7. Among these, gorilla-openfunctions-v2 represents one of the most advanced tool-finetuned models, while Yi-1.5-34B and deepseek-coder-33b are both 30B-level LLMs. Tool Decoding demonstrates significant improvements across all three models, with both deepseek-coder-33b and gorilla-openfunctions-v2 outperforming GPT-4.

## F  COMPARISON WITH EXISTING CONSTRAINED DECODING METHODS

There are two existing decoding methods for tool usage. In this section, we highlight how our method differs from them.

FANTASE (Wang et al., 2024) is not a plug-and-play method, as it requires additional training of a reranker. It introduces state-tracked constrained decoding to ensure the correct format but still relies on LLMs to generate all keys, including both required and optional parameters. This approach cannot effectively address issues like missing certain parameters, as illustrated in Figure 2 of Wang et al. (2024). To mitigate this limitation, a separate reranker should be trained to select the optimal tool call from multiple generated samples. In contrast, our method does not require any additional training and inherently avoids parameter absence, ensuring robustness in tool usage.

TOOlDEC (Zhang et al., 2023) is not universally applicable due to its specific requirements for tool call formats. It employs multiple Finite-State Machines (FSMs) to perform constrained decoding,

relying on a special token to signal transitions between FSMs as shown in Figure 4 of Zhang et al. (2023) . For instance, in their implementation, formats like `[Action:  ToolName, Action Input:  {key1=value1,<0x0A>key2=value2}]` are supported, where `<0x0A>` serves as an indicator for transition from the first value FSM to the next key FSM. Since values are generated freely, the model must independently generate this special token, which is then detected to trigger the FSM transition. This reliance introduces two key limitations: (1) If the model fails to adhere to the predefined format and omits the required special token during value generation, it remains stuck in the value mode, freely generating tokens. This disrupts the FSM transitions, rendering constrained decoding ineffective. (2) For tool-finetuned models or code models, such specialized formats may deviate from the data encountered during their fine-tuning or pretraining, potentially resulting in decreased performance.

It is important to note that punctuation marks, such as commas, spaces, and quotation marks, cannot serve as special tokens since most models encode them as part of surrounding tokens rather than as independent tokens. This makes TOOLDEC incompatible with common formats like `[ToolName(key1=value1, key2=value2)]`. In contrast, Tool Decoding determines transitions by verifying whether a complete variable of the specified type has been generated to assign the value, eliminating the reliance on special tokens. Table 6 demonstrates the robustness of our method compared to TOOLDEC across various tool-finetuned and code models.

## G  ADDITIONAL EXPERIMENT RESULTS

Due to space constraints, we present the detailed results in this section. Table 7 provides a comprehensive overview of the results from API-Bank, covering 7B-level models such as chat models, long-context models, code models, and lightweight models with 2B-level parameters. Table 8 displays the detailed results from BFCL V2 • Live. For brevity, weaker models that scored zero with both greedy search and beam search have been omitted.

Both Table 7 and Table 8 compare the performance of three decoding methods—greedy search, beam search, and Tool Decoding. The results consistently demonstrate that Tool Decoding significantly outperforms the other two methods across all evaluated models and benchmarks, with performance improvements exceeding twofold on certain tasks. This substantial enhancement highlights the effectiveness of Tool Decoding in addressing the limitations of greedy search and beam search, particularly in complex tool-use scenarios. Finally, Table 9 presents a comparison of the models' performance with and without order consistency under Tool Decoding on both benchmarks, further illustrating the effectiveness of our method in improving tool-use accuracy.

| Model | Decoding Method | Call | Retrieve+Call | Total |
|---|---|---|---|---|
| Closed-Source Models | | | | |
| GPT-4 | Greedy Search | 76.2 | 47.4 | 61.8 |
| GPT-3.5 | Greedy Search | 66.7 | 46.7 | 56.7 |
| Generalist Models | | | | |
| Mistral-7b-v0.1 | **Tool Decoding** | **65.7** | **50.4** | **58.1** |
| | Greedy Search | 31.3 | 31.9 | 31.6 |
| | Beam Search | 35.3 | 26.7 | 31.0 |
| FILM-7b | **Tool Decoding** | **70.4** | **52.6** | **61.5** |
| | Greedy Search | 37.3 | 43.7 | 40.5 |
| | Beam Search | 35.6 | 42.2 | 38.9 |
| deepseek-coder-6.7b | **Tool Decoding** | **70.9** | **55.6** | **63.3** |
| | Greedy Search | 46.9 | 43.0 | 45.0 |
| | Beam Search | 48.4 | 34.1 | 41.3 |
| gemma-7b | **Tool Decoding** | **67.4** | **46.7** | **57.1** |
| | Greedy Search | 53.9 | 34.8 | 44.4 |
| | Beam Search | 55.1 | 26.7 | 40.9 |
| Llama3-8b | **Tool Decoding** | **54.4** | **48.9** | **51.7** |
| | Greedy Search | 33.1 | 45.2 | 39.2 |
| | Beam Search | 34.6 | 32.6 | 33.6 |
| Qwen2-7b | **Tool Decoding** | **53.9** | **46.7** | **50.3** |
| | Greedy Search | 33.1 | 34.1 | 33.6 |
| | Beam Search | 32.8 | 38.5 | 35.7 |
| Yi-1.5-6b | **Tool Decoding** | **50.4** | **49.6** | **50.0** |
| | Greedy Search | 33.1 | 28.6 | 30.9 |
| | Beam Search | 38.9 | 21.9 | 30.4 |
| Yi-1.5-6b-Chat | **Tool Decoding** | **27.8** | **26.7** | **27.3** |
| | Greedy Search | 21.6 | 21.5 | 21.6 |
| | Beam Search | 19.3 | 18.5 | 18.9 |
| Yi-Coder-1.5b | **Tool Decoding** | **49.9** | **44.4** | **47.2** |
| | Greedy Search | 39.9 | 13.3 | 26.6 |
| | Beam Search | 41.9 | 11.9 | 26.9 |
| Yi-1.5-34B | **Tool Decoding** | **68.9** | **53.3** | **61.1** |
| | Greedy Search | 60.4 | 45.2 | 52.8 |
| deepseek-coder-33b | **Tool Decoding** | **74.4** | **57.0** | **65.7** |
| | Greedy Search | 57.9 | 46.7 | 52.3 |
| Tool-Finetuned Models | | | | |
| gorilla-openfunctions-v2 | **Tool Decoding** | **77.2** | **51.9** | **64.6** |
| | Greedy Search | 51.9 | 48.9 | 50.4 |
| | Beam Search | 48.4 | 45.2 | 46.8 |
| xLAM-7b-r | **Tool Decoding** | **73.9** | **54.8** | **64.4** |
| | Greedy Search | 36.1 | 41.5 | 38.8 |
| | Beam Search | 32.3 | 41.9 | 37.1 |
| Toolformer | **Tool Decoding** | **31.8** | **27.4** | **29.6** |
| | Greedy Search | 13.5 | 4.4 | 8.9 |
| | Beam Search | 23.3 | 8.2 | 15.8 |

Table 7: Detailed results on API-Bank, evaluated across a wide range of models.

| Model | Decoding Method | Simple | Multiple | Parallel | Parallel Multiple | Total |
|---|---|---|---|---|---|---|
| Closed-Source Models | | | | | | |
| GPT-4 | Greedy Search | 68.2 | 76.4 | 81.3 | 58.3 | 71.1 |
| GPT-3.5 | Greedy Search | 54.3 | 57.5 | 62.5 | 41.7 | 54.0 |
| Generalist Models | | | | | | |
| Mistral-7b-v0.1 | **Tool Decoding** | **57.0** | **41.9** | **43.8** | **41.7** | **46.1** |
| | Greedy Search | 15.9 | 18.9 | 12.5 | 0.0 | 11.8 |
| | Beam Search | 14.7 | 19.3 | 18.8 | 0.0 | 13.2 |
| FILM-7b | **Tool Decoding** | **64.3** | **69.1** | **62.5** | **33.3** | **57.3** |
| | Greedy Search | 53.1 | 61.4 | 0.0 | 8.3 | 30.7 |
| | Beam Search | 50.4 | 57.0 | 0.0 | 8.3 | 28.9 |
| deepseek-coder-6.7b | **Tool Decoding** | **65.9** | **55.8** | **68.8** | **58.3** | **62.2** |
| | Greedy Search | 23.3 | 1.1 | 12.5 | 4.2 | 10.3 |
| | Beam Search | 21.3 | 7.4 | 18.8 | 12.5 | 15.0 |
| Yi-Coder-1.5b | **Tool Decoding** | **52.3** | **28.0** | **37.5** | **16.7** | **33.6** |
| | Greedy Search | 0.0 | 0.0 | 0.0 | 0.0 | 0.0 |
| | Beam Search | 0.39 | 0.0 | 0.0 | 0.0 | 0.1 |
| Yi-1.5-6b | **Tool Decoding** | **61.6** | **38.1** | **50.0** | **41.7** | **47.9** |
| | Greedy Search | 0.0 | 0.0 | 0.0 | 0.0 | 0.0 |
| | Beam Search | 0.0 | 0.0 | 0.0 | 0.0 | 0.0 |
| Qwen2-7b | **Tool Decoding** | **58.1** | **60.7** | **50.0** | **45.8** | **53.7** |
| | Greedy Search | 44.2 | 34.3 | 12.5 | 29.2 | 30.1 |
| | Beam Search | 42.6 | 34.6 | 6.3 | 33.3 | 29.2 |
| Tool-Finetuned Models | | | | | | |
| xLAM-7b-r | **Tool Decoding** | **69.8** | **67.7** | **56.3** | **75.0** | **67.2** |
| | Greedy Search | 42.1 | 29.1 | 31.3 | 16.7 | 29.8 |
| | Beam Search | 40.7 | 33.9 | 31.3 | 20.8 | 31.68 |

Table 8: Detailed results on the BFCL V2 • Live, evaluated across a wide range of models. Some weak models that scored nearly zero with both greedy search and beam search have been omitted for brevity.

| Model | Decoding Method | API-Bank | BFCL V2 • Live |
|---|---|---|---|
| Generalist Models | | | |
| Mistral-7b-v0.1 | Tool Decoding w/ oc | 58.1 | 46.1 |
| | Tool Decoding w/o oc | 56.2 | 43.4 |
| FILM-7b | Tool Decoding w/ oc | 61.2 | 57.3 |
| | Tool Decoding w/o oc | 59.5 | 54.7 |
| deepseek-coder-6.7b | Tool Decoding w/ oc | 63.0 | 62.2 |
| | Tool Decoding w/o oc | 61.4 | 58.7 |
| Yi-Coder-1.5b | Tool Decoding w/ oc | 47.2 | 33.6 |
| | Tool Decoding w/o oc | 45.3 | 32.1 |
| Yi-1.5-6b | Tool Decoding w/ oc | 50.0 | 47.9 |
| | Tool Decoding w/o oc | 48.1 | 44.7 |
| Qwen2-7b | Tool Decoding w/ oc | 50.3 | 53.7 |
| | Tool Decoding w/o oc | 48.2 | 50.6 |
| Tool-Finetuned Models | | | |
| xLAM-7b-r | Tool Decoding w/ oc | 64.4 | 67.2 |
| | Tool Decoding w/o oc | 62.7 | 64.1 |

Table 9: Comparison of tool decoding with and without order consistency.

