# OpenReview forum: "Tool Decoding: A Plug-and-Play Approach to Enhancing Language Models for Tool Usage"
_ICLR.cc/2025/Conference — ICLR 2025 Conference Withdrawn Submission_

### Official Review · Reviewer_CYKv · 2024-10-30

**Soundness:** 3
**Presentation:** 3
**Contribution:** 3
**Rating:** 6
**Confidence:** 4

**Summary:**

The authors propose a constrained decoding workflow to improve LLMs' utilization of external tools. LLMs need to recognize when to enter tool mode, which tool to select, and how to invoke the tool with the appropriate arguments. Errors in any of these steps can result in an incorrect response. To address this, the authors introduce a constrained decoding approach that restricts the output vocabulary to include only tool names when the LLM enters tool mode. After a tool is selected, multiple candidate arguments (key-value pairs) are generated by varying the parameter order, and majority voting is used to select the final value for each parameter. The tool is then called, and the LLM continues generating its response. The authors evaluate their tool decoding method on the API-Bank and BFCL datasets across several LLMs, finding that their approach outperforms standard greedy and beam search decoding techniques.

**Strengths:**

The main benefit of the proposed constrained tool decoding approach is that it is training-free, making it easier to integrate into existing inference pipelines.
Constrained decoding provides a significant accuracy boost over greedy and beam search methods in terms of tool usage.
The authors conduct an error analysis to examine the impact of different error types across various LLMs, helping to identify key bottlenecks.
Additionally, the idea of generating multiple values for each parameter by varying the order is a compelling approach.

**Weaknesses:**

The proposed constrained decoding method increases latency, so it would have been beneficial to include a comparison in terms of latency impact. Additionally, incorporating some naive baselines, such as those using alternative search strategies like epsilon sampling or prompt engineering, would add value. As noted in the related work section, other constrained decoding algorithms are available, so a comparison to existing methods would have been interesting. Furthermore, the absence of stronger instruction-tuned LLMs as base models is a limitation.

**Questions:**

Could you please address the below questions:
1. Why UltraTool is used in Figure 2, while API-Bank is used in Figure 3?
2. Why the awareness error changes between the configurations with and without tool decoding in Figure 6?
3. Why LLM is used to generate optional parameters instead of supplying them directly, as is done with required parameters?
4. Provide a more detailed explanation on how accuracy is computed in the paper?
5. Report the success rate with and without constrained decoding, specifically indicating the proportion of samples with correct awareness, selection, and tool call?

Additional comments:
1. Figure 5 is missing accuracy numbers for Gemma.
2. Reporting accuracy numbers in Table 3 rather than error reduction might provide easier interpretation.
3. “Notably, tool call presents the greatest complexity, with even powerful models like Qwen1.5-72B achieving less than 60% accuracy in this stage” is a bit misleading because just by reading the text it appears that none of the model is able to achieve greater than 60% accuracy, which is not true because 32B model achieve around 70% accuracy. Maybe you can consider re-phrasing it to “Notably, tool call presents the greatest complexity, with the best performing model achieving around X% ”

---

> ### Author Response · Authors · 2024-11-23
> **Response to Reviewer CYKv (Part 1)**
>
> We thank Reviewer CYKv for providing constructive comments for our work. We will address your concerns in the following points.
>
> ---
>
> **W 1.** Latency analysis
>
> Thank you for your constructive advice! We have added latency analysis in **Appendix C**.
>
> The Table below shows the inference speed (sec/sample) for Tool Decoding  in comparison with greedy search and beam search. Tool Decoding with $oc \leq 1$ introduces only slight latency, while Tool Decoding with $oc \leq 6$ is faster than beam search with the same number of samples. This is because order consistency maintains multiple candidates only during the generation of the tool call, which constitutes just a portion of the entire response. Note that our current implementation applies constraints at the level of logits. For practical deployment, these constraints could be implemented at the language head layer, which would further reduce computational requirements and enhance processing speed.
>
> |        Model        | Greedy Search | Tool Decoding ($oc \leq 1$) | Beam Search ($beam=6$) | Tool Decoding ($oc \leq 6$) |
> | :-----------------: | :-----------: | :-------------------------: | :--------------------: | :-------------------------: |
> |   Mistral-7b-v0.1   |     7.92      |            8.57             |         13.82          |            9.74             |
> | deepseek-coder-6.7b |     7.69      |            8.41             |         15.25          |            10.7             |
>
>
> ---
>
> **W 2.** Conparison with other search strategies and prompt engineering
>
> **Epsilon sampling**
>
> Following your suggestion, we perform experiments for epsilon sampling on API-Bank (Call). The results show in the Table below. Since the performance of epsilon sampling is similar to greedy search, we decided not to include it as a baseline.
>
> |                     | $\epsilon=0.01$ | $\epsilon=0.001$ | greedy search |
> | :-----------------: | :-------------: | :--------------: | :-----------: |
> |   Mistral-7b-v0.1   |      32.8       |       31.6       |     31.3      |
> | deepseek-coder-6.7b |      41.4       |       40.1       |     46.9      |
>
> **Prompt engineering**
>
> As a plug-and-play method, Tool Decoding integrates seamlessly with prompt engineering. Table below presents the performance of different numbers of in-context examples on API-Bank (Call). The results demonstrate that our method effectively combines with prompt engineering, significantly enhancing the model’s tool usage capabilities. Notably, this combination even enables a 7B-level generalist model, such as deepseek-coder-6.7b, to surpass GPT-4 under the same prompt settings, as highlighted in **Bold values**. The results have been added in the **Section 4**.
>
> |      Model | Decoding Methods       | icl0 |   icl2   |   Icl4   |   Icl6   |   Icl8   |
> | :--------------:|:-------------------: | :--: | :------: | :------: | :------: | :------: |
> |       GPT4 | Greedy Search        | 76.2 |   72.7   |   72.2   |   73.7   |   73.4   |
> |   Mistral-7b-v0.1 | Greedy Search   | 31.3 |   47.1   |   45.1   |   50.4   |   43.6   |
> |   Mistral-7b-v0.1 | **Tool Decoding** | 65.7 |   70.2   |   69.2   |   70.5   |   70.9   |
> | deepseek-coder-6.7b | Greedy Search | 46.9 |   66.7   |   69.2   |   69.2   |   70.2   |
> | deepseek-coder-6.7b | **Tool Decoding** | 70.9 | **74.4** | **76.7** | **76.9** | **77.4** |

---

> ### Author Response · Authors · 2024-11-23
> **Response to Reviewer CYKv (Part 2)**
>
> **W 3.** Conparison with other constrained decoding algorithms
>
> There are two existing constrained decoding methods for tool usage. We have compared them with our method, and the results have been included in **Appendix F**.
>
> **FANTASE** [1] is not a plug-and-play method, as it requires additional training of a reranker. It introduces state-tracked constrained decoding to ensure the correct format but still relies on LLMs to generate all keys, including both required and optional parameters. This approach cannot effectively address issues like missing certain parameters, as illustrated in Figure 2 of [1]. To mitigate this limitation, a separate reranker should be trained to select the optimal tool call from multiple generated samples.  In contrast, our method does not require any additional training and inherently avoids parameter absence, ensuring robustness in tool usage. Considering that this approach is not training-free and the code has not been open-sourced, we did not include it in our experimental comparisons.
>
> **TOOlDEC** [2] is not universally applicable due to its specific requirements for tool call formats. It employs multiple Finite-State Machines (FSMs) to perform constrained decoding, relying on a special token to signal transitions between FSMs as shown in Figure 4 of [2] . For instance, in their implementation, formats like *[Action: ToolName, Action Input: \{key1=value1,<0x0A>key2=value2\}]* are supported, where *<0x0A>* serves as an indicator for transition from the first value FSM to the next key FSM. Since values are generated freely, the model must independently generate this special token, which is then detected to trigger the FSM transition. This reliance introduces two key limitations:
>
> 1. If the model fails to adhere to the predefined format and omits the required special token during value generation, it remains stuck in the value mode, freely generating tokens. This disrupts the FSM transitions, rendering constrained decoding ineffective.
> 2. For tool-finetuned models or code models, such specialized formats may deviate from the data encountered during their fine-tuning or pretraining, potentially resulting in decreased performance.
>
> It is important to note that punctuation marks, such as commas, spaces, and quotation marks, cannot serve as special tokens since most models encode them as part of surrounding tokens rather than as independent tokens. This makes TOOLDEC incompatible with common formats like *[ToolName(key1=value1, key2=value2)]*. In contrast, Tool Decoding determines transitions by verifying whether a complete variable of the specified type has been generated to assign the value, eliminating the reliance on special tokens. Table below demonstrates the robustness of our method compared to TOOLDEC across various tool-finetuned and code models.
>
> |                          | Greedy Search | Tool Decoding | TOOLDEC |
> | :----------------------: | :-----------: | :-----------: | :-----: |
> | gorilla-openfunctions-v2 |     51.9      |   **77.2**    |  69.4   |
> |        Toolformer        |     13.5      |   **31.8**    |   7.7   |
> |   deepseek-coder-6.7b    |     46.9      |   **70.9**    |  65.7   |
>
> [1] Zhuoer Wang, et al. "Fantastic sequences and
> where to find them: Faithful and efficient API call generation through state-tracked constrained decoding and reranking." In Findings of the Association for Computational Linguistics: EMNLP, 2024.
>
> [2] Kexun Zhang, et al. "Syntax error-free and generalizable tool use for llms via finite-state decoding." arXiv preprint arXiv:2310.07075, 2023

---

> ### Author Response · Authors · 2024-11-23
> **Response to Reviewer CYKv (Part 3)**
>
> **W 3.** Take stronger models as base models
>
> We supply results with three stronger models, including one powerful tool-finetuned models and two 30B-level models.
>
> In Table below, gorilla-openfunctions-v2 represents one of the most advanced tool-finetuned models, while Yi-1.5-34B and deepseek-coder-33b are both 30B-level LLMs. Tool Decoding demonstrates significant improvements across all three models, with both deepseek-coder-33b and gorilla-openfunctions-v2 outperforming GPT-4 on API-Bank. The results have been added in **Appendix E**.
>
> |                          |  Decoding Method  |   Call   | Retrieve+Call |  Total   |
> | :----------------------: | :---------------: | :------: | :-----------: | :------: |
> |          GPT-4           |   Greedy Search   |   76.2   |     47.4      |   61.8   |
> | gorilla-openfunctions-v2 |   Greedy Search   |   51.9   |     48.9      |   50.4   |
> | gorilla-openfunctions-v2 |    Beam Search    |   48.4   |     45.2      |   46.8   |
> | gorilla-openfunctions-v2 | **Tool Decoding** | **77.2** |   **51.9**    | **64.6** |
> |        Yi-1.5-34B        |   Greedy Search   |   60.4   |     45.2      |   52.8   |
> |        Yi-1.5-34B        | **Tool Decoding** | **68.9** |   **53.3**    | **61.1** |
> |    deepseek-coder-33b    |   Greedy Search   |   57.9   |     46.7      |   52.3   |
> |    deepseek-coder-33b    | **Tool Decoding** | **74.4** |   **57.0**    | **65.7** |
>
> ---
>
> **Q 1.** Why UltraTool is used in Figure 2, while API-Bank is used in Figure 3?
>
> UltraTool is a dataset specifically designed to provide fine-grained testing for the three distinct tool-use stages individually, so we utilize it to conduct a preliminary experiment to evaluate the performance of LLMs at each stage in isolation.
>
> However, tool usage is a continuous process across these stages rather than a series of isolated steps. Therefore, we conduct an error analysis using API-Bank, a widely used benchmark for tool-use dialogue, to better reflect the practical scenarios.
>
> ---
>
> **Q 2.** Why the awareness error changes between the configurations with and without tool decoding in Figure 6?
>
> Thank you for helping us identify a small bug in our error analysis. Awareness errors occur when the models fail to output the start signal for using a tool, such as [. However, we detected these errors by checking for the simultaneous absence of both [ and ], which caused a few cases to be incorrectly classified as awareness errors. We have fixed this bug and updated the corresponding results in the paper. The change is tiny and has no influence to our conclusion.
>
> ---
>
> **Q 3.** Why LLM is used to generate optional parameters instead of supplying them directly, as is done with required parameters?
>
> Required parameters are essential for invoking a tool, so we can directly supply them to ensure none are missing. However, optional parameters can be absent in a tool call. Whether to use them depends on the specific context, so we allow LLMs to make this decision autonomously through constrained decoding.
>
> ---
>
> **Q 4.** Provide a more detailed explanation on how accuracy is computed in the paper?
>
> The accuracy is computed following these steps:
>
> 1.	Extract the tool call from the generated content according to the predefined format.
> 2.	Verify whether the tool name generated is correct.
> 3.	Check whether all the keys used are correct.
> 4.	Confirm whether all the values used are correct.
>
> A tool usage is considered successful only if it passes all the above steps. Otherwise, it is classified as a failure.
>
> ---
>
> **Q 5.** Report the success rate with and without constrained decoding, specifically indicating the proportion of samples with correct awareness, selection, and tool call?
>
> We apologize for not fully understanding your points. We have reported our success rate and error distribution across three stages with and without Tool Decoding. Please refer to Figure 5 and 6 in paper.
>
> ---
>
> **AC 1.** Figure 5 is missing accuracy numbers for Gemma.
>
> The accuracy numbers for Gemma are provided in Appendix G. Due to space limitations, we have reported only a subset of the results in the main text.
>
> ---
>
> **AC 2.** Reporting accuracy numbers in Table 3 rather than error reduction might provide easier interpretation.
>
> Thanks for your advice. We aim to emphasize the performance of Order Consistency in mitigating value errors. Additionally, since the overall distribution of error types varies across models, using the proportion of value errors,  provides a clearer and more consistent basis for comparison across different models.
>
> ---
>
> **AC 3.** re-phrase in section 2.
>
> Thanks for your suggestion! We have re-phrased it in **Section 2** according to your comment.

---

> > ### Comment · Reviewer_CYKv · 2024-11-24
> >
> > Thank you authors for providing a detailed response and thorough clarifications.

---

> ### Author Response · Authors · 2024-11-26
>
> Thank you for your feedback! We are pleased to know that our responses resolved your concerns. If there are any further questions or suggestions, please don’t hesitate to let us know. Have a great day!

---

### Official Review · Reviewer_fh4s · 2024-10-30

**Soundness:** 2
**Presentation:** 3
**Contribution:** 3
**Rating:** 6
**Confidence:** 4

**Summary:**

This paper introduces a method to improve LLMs' tool-use capabilities with constrained decoding. They claim that current LLMs struggle with tool awareness, tool selection, and tool call. To address these, the author uses constrained decoding to reduce selection and format errors, moreover they also use order consistency to enhance parameter accuracy with structured sampling and majority voting.
This plug-and-play approach enables seamless adaptation to new tools only with the api documentation.
Experimental results on API-Bank and BFCL V2 Live benchmarks show high performance improvements across different models.
This work demonstrates the effectiveness of specialized decoding methods tailored to tool usage.

**Strengths:**

1. This work introduces an innovative, training-free approach to address tool-use challenges in language models. With constrained decoding and order consistency, the LLM can have much better performance over the two benchmark.
2. The paper is well-organized and uses clear figure to present the ideas, like constrained decoding. The figures break down workflows and methods, which helps improve readability.
3. This method provide a way to improve the LLM tool ability without training. This method may be important across diverse applications, enabling more efficient and versatile LLM deployment in real-world settings.

**Weaknesses:**

1. The author only compares to standard methods like greedy and beam search. However, I believe the author should do more comparison with recent tool-use improvement methods, such as state-tracked constrained decoding and reranking method or other in-context learning methods.
2. The models involved in the paper are mostly under 7B, (only dicuss 7B > models in the Figure 2), yet discussion about 30B, or even 70B model is limited in the paper. I am wondering whether the method still works or how much the method will improve when the model gets larger.
3.  Since the method involves structured sampling and majority voting, maybe the author can discuss its computational efficiency compared to baseline. This would give a more picture of its practical trade-offs in real-time applications.

**Questions:**

See questions in Weaknesses.

---

> ### Author Response · Authors · 2024-11-23
> **Response to Reviewer fh4s (Part 1)**
>
> We thank Reviewer fh4s for providing constructive comments for our work. We will address your concerns in the following points. All of the results below have been added in our paper.
>
> ---
>
> **W 1.1** Comparison with other constrained decoding methods
>
> There are two existing constrained decoding methods for tool usage. We have compared them with our method, and the results have been included in **Appendix F**.
>
> **FANTASE** [1] is not a plug-and-play method, as it requires additional training of a reranker. It introduces state-tracked constrained decoding to ensure the correct format but still relies on LLMs to generate all keys, including both required and optional parameters. This approach cannot effectively address issues like missing certain parameters, as illustrated in Figure 2 of [1]. To mitigate this limitation, a separate reranker should be trained to select the optimal tool call from multiple generated samples.  In contrast, our method does not require any additional training and inherently avoids parameter absence, ensuring robustness in tool usage.  Considering that this approach is not training-free and the code has not been open-sourced, we did not include it in our experimental comparisons.
>
> **TOOlDEC** [2] is not universally applicable due to its specific requirements for tool call formats. It employs multiple Finite-State Machines (FSMs) to perform constrained decoding, relying on a special token to signal transitions between FSMs as shown in Figure 4 of [2] . For instance, in their implementation, formats like *[Action: ToolName, Action Input: \{key1=value1,<0x0A>key2=value2\}]* are supported, where *<0x0A>* serves as an indicator for transition from the first value FSM to the next key FSM. Since values are generated freely, the model must independently generate this special token, which is then detected to trigger the FSM transition. This reliance introduces two key limitations:
>
> 1. If the model fails to adhere to the predefined format and omits the required special token during value generation, it remains stuck in the value mode, freely generating tokens. This disrupts the FSM transitions, rendering constrained decoding ineffective.
> 2. For tool-finetuned models or code models, such specialized formats may deviate from the data encountered during their fine-tuning or pretraining, potentially resulting in decreased performance.
>
> It is important to note that punctuation marks, such as commas, spaces, and quotation marks, cannot serve as special tokens since most models encode them as part of surrounding tokens rather than as independent tokens. This makes TOOLDEC incompatible with common formats like *[ToolName(key1=value1, key2=value2)]*. In contrast, Tool Decoding determines transitions by verifying whether a complete variable of the specified type has been generated to assign the value, eliminating the reliance on special tokens. Table below demonstrates the robustness of our method compared to TOOLDEC across various tool-finetuned and code models.
>
> |                          | Greedy Search | Tool Decoding | TOOLDEC |
> | :----------------------: | :-----------: | :-----------: | :-----: |
> | gorilla-openfunctions-v2 |     51.9      |   **77.2**    |  69.4   |
> |        Toolformer        |     13.5      |   **31.8**    |   7.7   |
> |   deepseek-coder-6.7b    |     46.9      |   **70.9**    |  65.7   |
>
> [1] Zhuoer Wang, et al. "Fantastic sequences and
> where to find them: Faithful and efficient API call generation through state-tracked constrained decoding and reranking." In Findings of the Association for Computational Linguistics: EMNLP, 2024.
>
> [2] Kexun Zhang, et al. "Syntax error-free and generalizable tool use for llms via finite-state decoding." arXiv preprint arXiv:2310.07075, 2023

---

> ### Author Response · Authors · 2024-11-23
> **Response to Reviewer fh4s (Part 2)**
>
> **W 1.2** Comparison with in-context learning
>
> As a plug-and-play method, Tool Decoding integrates seamlessly with prompt engineering. Table below presents the performance of different numbers of in-context examples on API-Bank (Call). The results demonstrate that our method effectively combines with prompt engineering, significantly enhancing the model’s tool usage capabilities. Notably, this combination even enables a 7B-level generalist model, such as deepseek-coder-6.7b, to surpass GPT-4 under the same prompt settings, as highlighted in **Bold values**. The results have been added in the **Section 4**.
>
> |      Model | Decoding Methods       | icl0 |   icl2   |   Icl4   |   Icl6   |   Icl8   |
> | :--------------:|:-------------------: | :--: | :------: | :------: | :------: | :------: |
> |       GPT4 | Greedy Search        | 76.2 |   72.7   |   72.2   |   73.7   |   73.4   |
> |   Mistral-7b-v0.1 | Greedy Search   | 31.3 |   47.1   |   45.1   |   50.4   |   43.6   |
> |   Mistral-7b-v0.1 | **Tool Decoding** | 65.7 |   70.2   |   69.2   |   70.5   |   70.9   |
> | deepseek-coder-6.7b | Greedy Search | 46.9 |   66.7   |   69.2   |   69.2   |   70.2   |
> | deepseek-coder-6.7b | **Tool Decoding** | 70.9 | **74.4** | **76.7** | **76.9** | **77.4** |
>
> ---
>
> **W 2.** Larger models
>
> Following your suggestion, we conducted experiments on two 30B-level generalist models to enhance the comprehensiveness of our evaluation. We have added the results in **Appendix E**.
> In Table below, Tool Decoding demonstrates significant improvements across these 30B-level models, with deepseek-coder-33b even outperforming GPT-4 on API-Bank.
>
> |       Model        |  Decoding Method  |   Call   | Retrieve+Call |  Total   |
> | :----------------: | :---------------: | :------: | :-----------: | :------: |
> |       GPT-4        |   Greedy Search   |   76.2   |     47.4      |   61.8   |
> |     Yi-1.5-34B     |   Greedy Search   |   60.4   |     45.2      |   52.8   |
> |     Yi-1.5-34B     | **Tool Decoding** | **68.9** |   **53.3**    | **61.1** |
> | deepseek-coder-33b |   Greedy Search   |   57.9   |     46.7      |   52.3   |
> | deepseek-coder-33b | **Tool Decoding** | **74.4** |   **57.0**    | **65.7** |
>
> For 70B-level models, we regret that limited computational resources prevent us from conducting more extensive experiments since Tool Decoding require deploying models locally. However, we performed a fine-grained error analysis with cloud APIs on two 70B-level models: qwen2-72b-instruct, which achieved an accuracy of 71.4%, and qwen1.5-72b-chat, which achieved 69.9%. The Table below presents the error type distribution for both models. Format and value errors remain the most prevalent, so we believe our method still works on these powerful models. The analysis has been added in **Appendix D**.
>
> |       Model        | Awareness | Selection | Format | Key  | Value |
> | :----------------: | :-------: | :-------: | :----: | :--: | :---: |
> | qwen2-72b-instruct |    8.0    |    7.1    |  27.4  | 8.0  | 49.6  |
> |  qwen1.5-72b-chat  |   24.2    |    9.2    |  28.3  | 9.2  | 29.2  |
>
> ---
>
> **W 3.** Computational efficiency
>
> Thank you for your constructive advice! We have added analysis on computational efficiency in **Appendix C**.
>
> The Table below shows the inference speed ((sec/sample) for Tool Decoding  in comparison with greedy search and beam search. Tool Decoding with $oc \leq 1$ introduces only slight latency, while Tool Decoding with $oc \leq 6$ is faster than beam search with the same number of samples. This is because order consistency maintains multiple candidates only during the generation of the tool call, which constitutes just a portion of the entire response. Note that our current implementation applies constraints at the level of logits. For practical deployment, these constraints could be implemented at the language head layer, which would further reduce computational requirements and enhance processing speed.
>
> |        Model        | Greedy Search | Tool Decoding ($oc \leq 1$) | Beam Search ($beam=6$) | Tool Decoding ($oc \leq 6$) |
> | :-----------------: | :-----------: | :-------------------------: | :--------------------: | :-------------------------: |
> |   Mistral-7b-v0.1   |     7.92      |            8.57             |         13.82          |            9.74             |
> | deepseek-coder-6.7b |     7.69      |            8.41             |         15.25          |            10.7             |

---

> > ### Comment · Reviewer_fh4s · 2024-11-25
> >
> > Thanks for the responses from the authors, I am glad to raise my score

---

> ### Author Response · Authors · 2024-11-26
>
> Thank you for raising the score! We are delighted to know that our responses addressed your concerns. If you have any additional questions or feedback, please feel free to let us know. Wishing you a wonderful day!

---

### Official Review · Reviewer_hS7t · 2024-11-02

**Soundness:** 3
**Presentation:** 4
**Contribution:** 3
**Rating:** 6
**Confidence:** 4

**Summary:**

The paper presents Tool Decoding, a training-free approach to improve LLMs’ tool-use capabilities, addressing common errors in tool awareness, tool selection, and tool call. By using constrained decoding for format accuracy and order consistency for parameter correctness, Tool Decoding significantly boosts tool-augmented performance on benchmarks e.g. API-Bank and BFCL V2 • Live, even surpassing GPT-4 in some cases. This adaptable approach enhances LLMs' effectiveness in real-world tool use without requiring additional training.

**Strengths:**

1. The paper presents Tool Decoding a novel, training-free approach to enhancing large language models (LLMs) in tool use. Unlike conventional methods relying on extensive fine-tuning, this plug-and-play solution integrates tool-specific data into the decoding phase. Tool decoding approach includes constrained decoding and order consistency modules, which is shown to effectively mitigate different tool errors e.g. Key error, Value error, and Format error.

2. The paper conducts an error analysis across three stages of tool usage—Tool Awareness, Tool Selection, and Tool Call. The analysis is done, using both API-Bank and BFCL V2 • Live datasets, which are benchmark datasets well-suited to assessing tool-augmented LLMs.

3. They showed the effectiveness of Tool Decoding by applying it to a variety of both general-purpose and tool-specialized models, evaluating them on the API-Bank and BFCL V2 • Live benchmarks. The experimental results show that Tool Decoding substantially improves performance across all models. Nearly all models achieve performance gains above 70% on both benchmarks. Among the 7B parameter models, five surpass GPT-3.5 in critical tasks, with two even outperforming GPT-4.

**Weaknesses:**

1. Limitation of Novelty as a long paper: constrained decoding for tool usage in LLMs has been employed in several prior works (e.g., [Domino](https://arxiv.org/abs/2403.06988): is proposed to optimize for general constrained text generation tasks with a focus on grammar and token alignment, [TOOLDEC](https://arxiv.org/pdf/2310.07075): eliminates syntax errors by constraining token choices using FSMs, focusing on maintaining tool syntax), and the specific approach to constrained decoding in this paper does not differ significantly.  As a result, the primary novelty of this paper lies in the order consistency mechanism, which offers only a limited improvement (Table 6).

2. Missing Baselines: The tool-finetuned models presented in the results table are outdated, and more recent, more powerful models (e.g., [Hermes Function Calling](https://github.com/NousResearch/Hermes-Function-Calling) and [Gorilla](https://github.com/ShishirPatil/gorilla)) are not included for comparison.

**Questions:**

1. Can you please further describe the differences of your approach (specially constrained decoding module) to previous work on constraint decoding for tool calling in LLMs?

2. Is there any technical reason that more recent tool-fine tuned models are not added for comparison?

---

> ### Author Response · Authors · 2024-11-23
> **Response to Reviewer hS7t (Part 1)**
>
> We thank Reviewer hS7t for providing constructive comments for our work. We will address your concerns in the following points. The results below have all been added in our paper.
>
> ---
>
> **W&Q 1.1.** Comparison with other constrained decoding methods
>
> There are two existing constrained decoding methods for tool usage. We have compared them with our method, and the results have been included in **Appendix F**.
>
> **FANTASE** [1] is not a plug-and-play method, as it requires additional training of a reranker. It introduces state-tracked constrained decoding to ensure the correct format but still relies on LLMs to generate all keys, including both required and optional parameters. This approach cannot effectively address issues like missing certain parameters, as illustrated in Figure 2 of [1]. To mitigate this limitation, a separate reranker should be trained to select the optimal tool call from multiple generated samples.  In contrast, our method does not require any additional training and inherently avoids parameter absence, ensuring robustness in tool usage. Considering that this approach is not training-free and the code has not been open-sourced, we did not include it in our experimental comparisons.
>
> **TOOlDEC** [2] is not universally applicable due to its specific requirements for tool call formats. It employs multiple Finite-State Machines (FSMs) to perform constrained decoding, relying on a special token to signal transitions between FSMs as shown in Figure 4 of [2] . For instance, in their implementation, formats like *[Action: ToolName, Action Input: \{key1=value1,<0x0A>key2=value2\}]* are supported, where *<0x0A>* serves as an indicator for transition from the first value FSM to the next key FSM. Since values are generated freely, the model must independently generate this special token, which is then detected to trigger the FSM transition. This reliance introduces two key limitations:
>
> 1. If the model fails to adhere to the predefined format and omits the required special token during value generation, it remains stuck in the value mode, freely generating tokens. This disrupts the FSM transitions, rendering constrained decoding ineffective.
> 2. For tool-finetuned models or code models, such specialized formats may deviate from the data encountered during their fine-tuning or pretraining, potentially resulting in decreased performance.
>
> It is important to note that punctuation marks, such as commas, spaces, and quotation marks, cannot serve as special tokens since most models encode them as part of surrounding tokens rather than as independent tokens. This makes TOOLDEC incompatible with common formats like *[ToolName(key1=value1, key2=value2)]*. In contrast, Tool Decoding determines transitions by verifying whether a complete variable of the specified type has been generated to assign the value, eliminating the reliance on special tokens. Table below demonstrates the robustness of our method compared to TOOLDEC across various tool-finetuned and code models.
> |                          | Greedy Search | Tool Decoding | TOOLDEC |
> | :----------------------: | :-----------: | :-----------: | :-----: |
> | gorilla-openfunctions-v2 |     51.9      |   **77.2**    |  69.4   |
> |        Toolformer        |     13.5      |   **31.8**    |   7.7   |
> |   deepseek-coder-6.7b    |     46.9      |   **70.9**    |  65.7   |
>
> [1] Zhuoer Wang, et al. "Fantastic sequences and
> where to find them: Faithful and efficient API call generation through state-tracked constrained decoding and reranking." In Findings of the Association for Computational Linguistics: EMNLP, 2024.
>
> [2] Kexun Zhang, et al. "Syntax error-free and generalizable tool use for llms via finite-state decoding." arXiv preprint arXiv:2310.07075, 2023
>
> ---
>
> **W 1.2.** Limitation of Novelty
>
> Thanks for your comments, but we believe that our work introduces sufficient contributions and novelty . From a methodological perspective,  although a few works have introduced constrained decoding for tool usage, our method stands out as it is training-free and can be better applied  to a broader range of models, as demonstrated in the comparisons above. Additionally, we propose order consistency, which can further enhance performance, building on the significant improvements achieved by constrained decoding. From an experimental perspective, we conduct detailed analyses and comprehensive experiments across a wide range of models, achieving performance gains exceeding 70% for nearly all models, as shown in Table 7 and 8. As a plug-and-play method, our approach further enhances performance when combined with tool-finetuned models and prompt engineering, highlighting its flexibility and effectiveness, as shown Table 3 and 7.

---

> ### Author Response · Authors · 2024-11-23
> **Response to Reviewer hS7t (Part 2)**
>
> **W&Q 2.** More powerful models.
>
> Following your suggestion, we conducted experiments on more powerful tool-finetuned models, such as gorilla-openfunctions-v2. Additionally, we included experiments on two 30B-level generalist models to enhance the comprehensiveness of our evaluation. The results have been added in **Appendix E**.
> In Table below, Tool Decoding demonstrates significant improvements across all three models, with both deepseek-coder-33b and gorilla-openfunctions-v2 outperforming GPT-4 on API-Bank.
>
> |                          |  Decoding Method  |   Call   | Retrieve+Call |  Total   |
> | :----------------------: | :---------------: | :------: | :-----------: | :------: |
> |          GPT-4           |   Greedy Search   |   76.2   |     47.4      |   61.8   |
> | gorilla-openfunctions-v2 |   Greedy Search   |   51.9   |     48.9      |   50.4   |
> | gorilla-openfunctions-v2 |    Beam Search    |   48.4   |     45.2      |   46.8   |
> | gorilla-openfunctions-v2 | **Tool Decoding** | **77.2** |   **51.9**    | **64.6** |
> |        Yi-1.5-34B        |   Greedy Search   |   60.4   |     45.2      |   52.8   |
> |        Yi-1.5-34B        | **Tool Decoding** | **68.9** |   **53.3**    | **61.1** |
> |    deepseek-coder-33b    |   Greedy Search   |   57.9   |     46.7      |   52.3   |
> |    deepseek-coder-33b    | **Tool Decoding** | **74.4** |   **57.0**    | **65.7** |

---

> ### Author Response · Authors · 2024-11-26
>
> Dear Reviewer hS7t,
>
> We have carefully prepared a response to address your concerns. Could you kindly take a moment to review it and let us know if it resolves the issues you raised? If you have any additional questions or suggestions, we would be happy to address them.
>
> Thank you for your time and consideration. Wishing you a great day!
>
> The Authors

---

### Official Review · Reviewer_PjPG · 2024-11-03

**Soundness:** 3
**Presentation:** 3
**Contribution:** 3
**Rating:** 6
**Confidence:** 2

**Summary:**

This paper addresses the limitations of LLMs in tool usage. Previous works often overlook the specific requirements of tool selection, format adherence, and parameter accuracy. The authors propose a training-free approach called "Tool Decoding," designed to improve tool-use precision through constrained decoding. Experimental results demonstrate that Tool Decoding achieves improvements in tool-use accuracy across benchmarks (API-Bank and BFCL V2), with open-weight models performing comparably to GPT-4 in some cases. However, the paper lacks a thorough comparison with other existing methods (only compared with beam and greedy search) and provides no report on latency, which could be important for practical applications.

**Strengths:**

- Achieved comparable results to GPT-4 for open-weight models with the help of Tool Decoding on the API-Bank and BFCL V2 benchmarks.
- The method does not require any training and primarily relies on constrained decoding to ensure format correctness and eliminate hallucinations.

**Weaknesses:**

- Comparison with other methods? The authors compare their method only with Beam Search and Greedy Search. Are there no other methods available, such as those in https://arxiv.org/pdf/2401.06201 (01.2024), trained models, etc.?
- For the error analysis section, I believe it does not sufficiently acknowledge that other researchers have reported similar problems. It is written as though this paper is the first to report them. For example, the API-Bank [paper](https://arxiv.org/pdf/2304.08244) provides a thorough analysis at a similar level of granularity in Section 7.3, which I believe has not been adequately acknowledged.

**Questions:**

- Value errors increased in most scenarios after applying Tool Decoding (add to Limitation section); on the other hand, key errors are not a major challenge. Tool Decoding primarily addresses format errors and partially resolves selection errors. I understand how it can help with selection errors by decoding tool names under constraints. However, for format errors, the example in Table 1 shows a missing parenthesis. How would Tool Decoding help in this case?

- Another important factor in decoding tools is latency in seconds, but I did not find any comparison. Although there is no major comparison with other methods, some latency figures for the tool itself would be beneficial.

---

> ### Author Response · Authors · 2024-11-23
> **Response to Reviewer PjPG (Part 1)**
>
> We thank Reviewer PjPG for providing constructive comments for our work. We will address your concerns in the following points.  The results below have all been added in our paper.
>
> ---
>
> **W 1.** Comparison with other methods.
>
> Thanks for your suggestion! Since Tool Decoding is a plug-and-play method, we can combine it with many other approaches such as prompt engineering or trained models.  We further expand our comparisons with more approaches as well as the  combination of them and our method. As shown in the following, Tool Decoding is **orthogonal to these methods** and can **largely improve the performance of them**. And the results are also added in **Section 4** and **Appendix E**.
>
> **In-context learning**
>
> As a plug-and-play method, Tool Decoding integrates seamlessly with prompt engineering. Table below presents the performance of different numbers of in-context examples on API-Bank (Call). The results demonstrate that our method effectively combines with prompt engineering, significantly enhancing the model’s tool usage capabilities. Notably, this combination even enables a 7B-level generalist model, such as deepseek-coder-6.7b, to surpass GPT-4 under the same prompt settings, as highlighted in **Bold values**.
>
> |      Model | Decoding Methods       | icl0 |   icl2   |   Icl4   |   Icl6   |   Icl8   |
> | :--------------:|:-------------------: | :--: | :------: | :------: | :------: | :------: |
> |       GPT4 | Greedy Search        | 76.2 |   72.7   |   72.2   |   73.7   |   73.4   |
> |   Mistral-7b-v0.1 | Greedy Search   | 31.3 |   47.1   |   45.1   |   50.4   |   43.6   |
> |   Mistral-7b-v0.1 | **Tool Decoding** | 65.7 |   70.2   |   69.2   |   70.5   |   70.9   |
> | deepseek-coder-6.7b | Greedy Search | 46.9 |   66.7   |   69.2   |   69.2   |   70.2   |
> | deepseek-coder-6.7b | **Tool Decoding** | 70.9 | **74.4** | **76.7** | **76.9** | **77.4** |
>
> **Trained models**
>
> Please note that we have already combined our method with some trained models and compared with them in Figure 5, Table 7 and Table 8 (xLAM-7b-r and Toolformer). We cite the results on API-Bank below.
>
> |           Model | Decoding Methods         |   Call   |   Retrieve+Call   |   Total   |
> | :-----------------:|:-------------------------: | :---: | :------: | :------: |
> |       xLAM-7b-r |**Tool Decoding**         | **73.9** | **54.8** | **64.4** |
> |       xLAM-7b-r |Greedy Search           | 36.1  |   41.5   |   38.8   |
> |       xLAM-7b-r |Beam Search              | 32.3  |   41.9   |   37.1   |
> |       Toolformer |**Tool Decoding**           | **31.8** | **27.4** | **29.6** |
> |       Toolformer |Greedy Search           | 13.5  |   4.4    |   8.9    |
> |       Toolformer |Beam Search              | 23.3  |   8.2    |   15.8   |
>
> In Table below, we also conduct further experiments on gorilla-openfunctions-v2, one of the most advanced trained models. Tool Decoding demonstrates significant improvements across all three models, with both xLAM-7b-r and gorilla-openfunctions-v2 outperforming GPT-4 on API-Bank.
>
> |          Model           |  Decoding Method  |   Call   | Retrieve+Call |  Total   |
> | :----------------------: | :---------------: | :------: | :-----------: | :------: |
> |          GPT-4           |   Greedy Search   |   76.2   |     47.4      |   61.8   |
> | gorilla-openfunctions-v2 | **Tool Decoding** | **77.2** |   **51.9**    | **64.6** |
> | gorilla-openfunctions-v2 |   Greedy Search   |   51.9   |     48.9      |   50.4   |
> | gorilla-openfunctions-v2 |    Beam Search    |   48.4   |     45.2      |   46.8   |
>
> ---
>
> **W 2.** Inadequate acknowledgment in the error analysis part.
>
> We apologize for this oversight. We have now adequately acknowledge it at the beginning of the Analysis of Errors part in **Section 2**: "While some existing works have conducted coarse error analyses, their evaluations are not sufficiently comprehensive and lack a systematic approach. For instance, the analysis in API-Bank (Li et al., 2023) overlooks value errors and includes ambiguous error types such as *Has Exception*, limiting both clarity and utility. In contrast, we conduct a stage-specific and comprehensive error analysis, systematically identifying errors at each stage to derive fine-grained insights."

---

> ### Author Response · Authors · 2024-11-23
> **Response to Reviewer PjPG (Part 2)**
>
> **Q1.1.** Value errors increased in most scenarios after applying Tool Decoding.
>
> The added value errors are not caused by Tool Decoding but are potential value errors previously hidden by format and key errors. Specifically, parameter values can only be extracted when the format and keys are correct. By addressing these format and key errors, Tool Decoding makes more values can be extracted and exposes these hidden value errors, leading to a slight increase in the observed proportion of value errors.
>
> In contrast, our method mitigates value errors: constrained decoding does not impact value generation, while order consistency actively reduces value errors, as demonstrated in Table 4.
>
> ---
>
> **Q1.2.**  For format errors, the example in Table 1 shows a missing parenthesis. How would Tool Decoding help in this case?
>
> Briefly speaking, we detect whether the last value has been fully generated, and once completed, the model is constrained to generate the closing parenthesis $)$.
>
> For example, suppose tool_A holds 2 required parameters, $\alpha$ and $\beta$, and has 1 optional parameters, $\gamma$. Once the last required parameter, $\beta$, is completely assigned, we constrain the model’s vocabulary space to [$\gamma,)$]. This leads to two possible scenarios:
>
> 1.	If the model outputs $)$, the tool call is considered complete.
> 2.	If the model outputs $\gamma$, the assignment process continues for $\gamma$. After the value of $\gamma$ is fully generated, since all parameters have been completed, we directly supply $)$ to finalize the tool call.
>
> Maybe your question is how we determine whether a value has been completely assigned. In expressions like [ToolName(key1=value1, key2=value2)], there is no special token explicitly indicating this completion. Most models encode punctuation marks, such as commas, spaces, and quote marks, as part of surrounding tokens. For instance, the expression "key1=**value1, key2**=value2" might be tokenized as [“key1”, “=”, **“value1,”**, **“ key2”**, “=”, “value2”].
>
> To address this, we determine whether a value has been completely assigned by checking if a full variable of the given type has been generated. Consequently, for Tool Decoding, handling the assignment of the last parameter is no different from handling the previous ones.
>
> ---
>
> **Q 2.** Latency analysis
>
> Thank you for your constructive advice! We supply the results below and exhibit the low latency of our method. The analysis has been added in **Appendix C**.
>
> The Table below shows the inference speed (sec/sample) for Tool Decoding  in comparison with greedy search and beam search. Tool Decoding with $oc \leq 1$ introduces only slight latency, while Tool Decoding with $oc \leq 6$ is faster than beam search with the same number of samples. This is because order consistency maintains multiple candidates only during the generation of the tool call, which constitutes just a portion of the entire response. Note that our current implementation applies constraints at the level of logits. For practical deployment, these constraints could be implemented at the language head layer, which would further reduce computational requirements and enhance processing speed.
>
> |        Model        | Greedy Search | Tool Decoding ($oc \leq 1$) | Beam Search ($beam=6$) | Tool Decoding ($oc \leq 6$) |
> | :-----------------: | :-----------: | :-------------------------: | :--------------------: | :-------------------------: |
> |   Mistral-7b-v0.1   |     7.92      |            8.57             |         13.82          |            9.74             |
> | deepseek-coder-6.7b |     7.69      |            8.41             |         15.25          |            10.7             |

---

> > ### Comment · Reviewer_PjPG · 2024-11-26
> >
> > I appreciate the resolution of inadequate acknowledgment to API-Bank and the addition of the latency analysis. I will raise my score to 6.

---

> ### Author Response · Authors · 2024-11-26
>
> Thank you for raising the score! We are delighted to know that our responses addressed your concerns. If you have any additional questions or feedback, please feel free to let us know. Wishing you a wonderful day!

---

### Author Response · Authors · 2024-11-23
**A summary of paper updates**

Following the reviewers’ suggestions, we have updated the paper with key revisions highlighted in orange. The main changes are as follows:
1. **Section 2:** Revised based on feedback from reviewer PjPG and reviewer CYKv.
2. **Section 4:** Presented results combining Tool Decoding with in-context learning.
3. **Appendix C:** Included latency analysis for our method.
4. **Appendix D:** Added error analysis for two 70B-level models.
5. **Appendix E:** Added results combining Tool Decoding with two 30B-level generalist models and one powerful tool-finetuned models.
6. **Appendix F:** Added comparisons with existing constrained decoding methods.

---

### Note · Authors · 2025-02-12

I have read and agree with the venue's withdrawal policy on behalf of myself and my co-authors.

---

### Meta-Review · Area_Chair_gZNH · 2025-01-03

**Metareview:**

This paper breaks down LLMs' tool usage into three stages - tool awareness, selection, and invocation - and identifies common failure modes at each step. The authors propose Tool Decoding, a simple constrained sampling approach that improves tool usage without extra training. The method restricts vocabulary during tool selection and uses majority voting across different parameter orderings during invocation. A key advantage is that it only needs API documentation to work with new tools. Experiments on API-Bank and BFCL V2 show substantial gains across different LLMs, sometimes matching GPT-4's performance.

In the initial round of reviews, the main criticism focused on the lack of comparative analysis. In the rebuttal, the authors expanded their discussion in Section 4 and Appendices E and F, adding comparisons with sampling methods like epsilon-sampling and with models fine-tuned for tool usage, including Gorilla-OpenFunctions, Toolformer, and DeepSeek-Coder. They also address reviewer fh4s’s concern by testing larger models in Appendix E. Another issue was the lack of latency analysis, which the authors have now fixed by providing detailed latency results in Appendix C.

After reviewing these revisions and engaging in follow-up discussions, all reviewers have agreed that the paper is suitable for acceptance.

**Additional Comments On Reviewer Discussion:**

See above.

---

### Decision · Program_Chairs · 2025-01-22

Accept (Poster)